# Scaling Offline RL via
# Efficient and Expressive Shortcut Models

**Nicolas Espinosa-Dice**
Cornell University
ne229@cornell.edu

**Yiyi Zhang**
Cornell University
yz2364@cornell.edu

**Yiding Chen**
Cornell University
yc2773@cornell.edu

**Bradley Guo**
Cornell University
bzg4@cornell.edu

**Owen Oertell**
Cornell University
ojo2@cornell.edu

**Gokul Swamy**
Carnegie Mellon University
gswamy@andrew.cmu.edu

**Kianté Brantley**
Harvard University
kdbrantley@harvard.edu

**Wen Sun**
Cornell University
ws455@cornell.edu

## Abstract

Diffusion and flow models have emerged as powerful generative approaches capable of modeling diverse and multimodal behavior. However, applying these models to offline reinforcement learning (RL) remains challenging due to the iterative nature of their noise sampling processes, making policy optimization difficult. In this paper, we introduce *Scalable Offline Reinforcement Learning* (SORL), a new offline RL algorithm that leverages shortcut models—a novel class of generative models—to scale both training and inference. SORL's policy can capture complex data distributions and can be trained simply and efficiently in a one-stage training procedure. At test time, SORL introduces both sequential and parallel inference scaling by using the learned $Q$-function as a verifier. We demonstrate that SORL achieves strong performance across a range of offline RL tasks and exhibits positive scaling behavior with increased test-time compute. We release the code at nico-espinosadice.github.io/projects/sorl.

## 1 Introduction

Offline reinforcement learning (RL) [Ernst et al., 2005, Lange et al., 2012, Levine et al., 2020] is a paradigm for using fixed datasets of interactions to train agents without online exploration. In this paper, we tackle the challenge of scaling offline RL, for which there are two core components: *training* and *inference*.

In order to scale training, offline RL algorithms must be capable of handling larger, more diverse multi-modal datasets [ONeill et al., 2024, Rafailov et al., 2024, Park et al., 2024a, Hussing et al., 2023, Gürtler et al., 2023, Bu et al., 2025]. While standard Gaussian-based policy classes cannot model multi-modal data distributions, flow matching [Lipman et al., 2022, Liu et al., 2022, Albergo et al., 2023] and diffusion models [Sohl-Dickstein et al., 2015, Ho et al., 2020, Song et al., 2021] have emerged as powerful, highly expressive model classes capable of modeling complex data distributions. However, while generative models like diffusion and flow matching can model diverse offline data, applying them to offline RL is challenging due to their iterative noise sampling process, which makes policy optimization difficult, often requiring backpropagation through time or distillation of a larger model. Ultimately, we desire a training procedure that can efficiently train an expressive model class.

39th Conference on Neural Information Processing Systems (NeurIPS 2025).

The second core component in scaling offline RL is inference. During inference, we desire both efficiency and precision: the agent must be able to make decisions rapidly (e.g. autonomous vehicles) but also take precise actions (e.g. surgical robots). Inspired by the recent work in test-time scaling of large language models [Wei et al., 2023, DeepSeek-AI, 2025, Gui et al., 2024, Brown et al., 2024, Muennighoff et al., 2025, Madaan et al., 2023, Qu et al., 2024, Qin et al., 2025], we investigate how test-time scaling can be applied in offline RL with generative models. That is, we desire an approach that can perform inference efficiently—avoiding the slow, many-step generation process of diffusion models—but can also leverage additional test-time compute when available.

Unfortunately, recent work in offline RL fails to achieve both of the desiderata necessary to scale offline RL. Distillation-based approaches to offline RL with generative models [Ding and Jin, 2023, Chen et al., 2023, 2024, Park et al., 2025] avoid extensive backpropagation through time during training. However, they may require more complex, two-stage training procedures (e.g. propagating through teacher/student networks), over which error compounds. They also have limited scaling of inference steps, thus losing expressivity when compared to multi-step generative models [Frans et al., 2024]. Alternatively, diffusion-based approaches that leverage backpropagation through time [Wang et al., 2022, He et al., 2023, Zhang et al., 2024, Ada et al., 2024] may learn policies with greater expressivity. However, diffusion models have slow inference, requiring a larger number of steps to generate high quality outputs [Frans et al., 2024].

At a high-level, offline RL algorithms that employ generative models struggle with the following tradeoff. In order to achieve training efficiency, we want to avoid performing many steps of the iterative noise sampling process during policy optimization, which often requires backpropagating many steps through time. However, modeling complex distributions in the offline data, such as diverse or multi-modal data, may require a larger number of discretization steps to allow for maximum expressivity. Finally, at inference-time, we desire the ability to both generate actions efficiently via a few-step sampling procedure (e.g. in robot locomotion settings) *and* leverage additional test-time compute when available (e.g. in robot manipulation settings).

In this paper, we tackle the question of how to achieve efficient training, while maintaining expressivity, *and* scale with greater inference-time compute. We introduce *Scalable Offline Reinforcement Learning* (SORL): a simple, efficient one-stage training procedure for expressive policies. Our key insight is to leverage *shortcut models* [Frans et al., 2024], a novel class of generative models, in order to incorporate *self-consistency* into the training process, thus allowing the policy to generate high quality samples under *any* inference budget. SORL's self consistency property allows us to vary the number of denoising steps used for the three core components in offline RL: policy optimization (i.e. the number of backpropagation through time steps), regularization to offline data (i.e. the total number of discretization steps), and inference (i.e. the number of inference steps). We can use fewer steps for policy optimization, thus making training efficient, while performing inference under varying inference budgets, depending on the desired inference-time compute budget. We incorporate shortcut models into regularized actor-critic algorithm and make the following contributions:

1. We introduce SORL**, an efficient, one-stage training procedure for expressive policies that can perform inference under *any* compute budget**, including one-step inference. Despite its efficiency, SORL's policy can capture complex, multi-modal data distributions.

2. Through a novel theoretical analysis of shortcut models, **we prove that SORL's training objective regularizes SORL's policy to the behavior of the offline data**.

3. Empirically, SORL **achieves the best performance against 10 baselines on a range of diverse tasks in offline RL**.

4. At test-time, SORL **can scale with greater inference-time compute by increasing the number of inference steps (i.e. sequential scaling) and performing best-of-$N$ sampling (i.e. parallel scaling)**. We show empirically that SORL can make up for a smaller training-time compute budget with a greater inference-time compute budget. SORL can also generalize to more inference steps than the number of steps optimized during training.

## 2   Background

**Markov Decision Process.**   We consider an infinite-horizon Markov Decision Process (MDP) [Puterman, 2014], $\mathcal{M} = \langle \mathcal{X}, \mathcal{A}, P, R, \gamma, \mu \rangle$. $\mathcal{X}$ and $\mathcal{A}$ are the state space and action space, respectively.

$P : \mathcal{X} \times \mathcal{A} \to \Delta(\mathcal{X})$ is the transition function, $R : \mathcal{X} \times \mathcal{A} \to [0, 1]$ is the reward function, and $\gamma$ is the discount factor. $\mu \in \Delta(\mathcal{X})$ is the starting state distribution. Let $\Pi = \{\pi : \mathcal{X} \to \Delta(\mathcal{A})\}$ be the class of stationary policies. We define the state-action value function $Q(x, a) : \mathcal{X} \times \mathcal{A} \to \mathbb{R}$ as $Q^\pi(x, a) := \mathbb{E}_\pi \left[ \sum_{i=1}^\infty R(x_i, a_i) \mid x_i = x, a_i = a \right]$. We define the state visitation distribution generated by a policy $\pi$ to be $d_\mu^\pi := (1 - \gamma)\mathbb{E}_{x_0 \sim \mu} \left[ \sum_{i=0}^\infty \gamma^i \Pr_i^\pi(x \mid x_0) \right]$. In the offline RL setting, we assume access to a fixed dataset $\mathcal{D} = \{(x, a, r, x')^{(j)}\}_{j \in 1,\ldots,n}$, such that the data was generated $(x, a, x') \sim d^{\pi_B}$ by some behavior policy $\pi_B$.

**Flow Matching.** We define flow matching [Lipman et al., 2022, Liu et al., 2022] as follows. Let $p^\star \in \Delta(\mathbb{R}^d)$ be the target data distribution in $d$-dimensional Euclidean space. Let $\bar{z}_0 \sim \mathcal{N}(0, I)$ and $\bar{z}_1 \sim p^\star$ be two independent random variables. We define $\bar{z}_t$ to be the linear-interpolation between the $\bar{z}_0$ and $\bar{z}_1$:

$$\bar{z}_t := t\bar{z}_1 + (1 - t)\bar{z}_0, \quad 0 \le t \le 1. \tag{1}$$

$\{\bar{z}_t\}_{t \in [0,1]}$ is fully determined by the start point $\bar{z}_0$ and end point $\bar{z}_1$. We use $\{p_t\}_{t \in [0,1]}$ to denote the sequence of marginal distributions of $\bar{z}_t$'s, where $p_0 = \mathcal{N}(0, I)$ and $p_1 = p^\star$.

The *drift function* $v_t(\cdot) : \mathbb{R}^d \to \mathbb{R}^d$ is defined to be the solution to the following least square regression:

$$\min_f \int_0^1 \mathbb{E}_{\bar{z}_0 \sim \mathcal{N}(0,I), \bar{z}_1 \sim p^\star} \left[ \|\bar{z}_1 - \bar{z}_0 - f(t\bar{z}_1 + (1 - t)\bar{z}_0, t)\|_2^2 \right] \mathrm{d}t. \tag{2}$$

We use $f^\star$ to denote the solution to the optimization problem in Equation 2 and define:

$$v_t(z) := f^\star(z, t), \quad \forall z \in \mathbb{R}^d, t \in [0, 1].$$

The drift function $v_t(\cdot)$ induces an ordinary differential equation (ODE)

$$\frac{\mathrm{d}}{\mathrm{d}t} z_t = v_t(z_t). \tag{3}$$

An appealing property of Equation 3 is: if the initial $z_0$ is drawn from Gaussian distribution $\mathcal{N}(0, I)$, then the marginal distribution of $z_t$, the solution to the ODE, is exactly $p_t$, the marginal distribution of the linear-interpolation process [Liu et al., 2022]. In practice, one can learn a drift function $\hat{v}_t(\cdot)$ by directly optimizing Equation 2, starting from $z_0 \sim \mathcal{N}(0, I)$, and solve the ODE: $\mathrm{d}z_t = \hat{v}_t(z_t)\mathrm{d}t$. The solution $z_1$ is an approximate sample from the target distribution $p^\star$.

**Shortcut Models.** Flow matching typically rely on small, incremental steps, which can be computationally expensive at inference time. Shortcut models improve inference efficiency by learning to take larger steps along the flow trajectory [Frans et al., 2024]. The key insight of Frans et al. [2024] is to condition the model not only on the timestep $t$, as in standard flow matching, but also on a step size $h$. The shortcut model $s(z_t, t, h)$ predicts the normalized direction from $z_t$ towards the correct next point $z_{t+h}$, such that

$$z_t + s(z_t, t, h)h \approx z_{t+h}. \tag{4}$$

The objective is to learn a shortcut model $s_\theta(z_t, t, h)$ for all combinations of $z_t, t, h$. The loss function is given by

$$\mathcal{L}^S(\theta) = \mathbb{E}_{\substack{z_0 \sim \mathcal{N}(0,I), \\ z_1 \sim D, (t,h) \sim p(t,h)}} \left[ \Big\| \underbrace{s_\theta(z_t, t, 1/M) - (z_1 - z_0)}_{\text{Flow Matching}} \Big\|^2 + \Big\| \underbrace{s_\theta(z_t, t, 2h) - s_{\text{target}}}_{\text{Self-Consistency}} \Big\|^2 \right], \tag{5}$$

where $s_{\text{target}} = s_\theta(z_t, t, h)/2 + s_\theta(z_{t+h}', t, h)/2$ and $z_{t+h}' = z_t + s_\theta(z_t, t, h)h$. $h$ is sampled uniformly from the reciprocals of powers of 2 between 1 and $M$, the maximum number of discretization steps, and $t$ is sampled uniformly between 0 and 1, so $p(h, t) = \text{Unif}\big(\{2^{-k}\}_{k=0}^{\lfloor \log_2 M \rfloor}\big) \times \text{Unif}(0, 1)$.[1]

The loss's first component is standard flow matching which ensures that when $h = 1/M$, the smallest step size, the model recovers the target direction $z_1 - z_0$. The second component, self-consistency, encourages the model to produce consistent behavior across step sizes. Intuitively, it ensures that one large *jump* of size $2h$ is equivalent to two combined *jumps* of size $h$. Thus, the model learns to take larger jumps with more efficient, fewer-step inference procedures [Frans et al., 2024].

---

[1] Note that the interval [0, 1] is discretized into $M$ steps.

**Algorithm 1:** Scalable Offline Reinforcement Learning (SORL)

**Data:** Offline dataset $\mathcal{D}$
**while** not converged **do**

Sample $(x, a^1, x', r) \sim \mathcal{D}$, $\quad a^0 \sim \mathcal{N}(0, I)$, $\quad (h, t) \sim p(h, t)$    # Parallelize batch
$a^t \leftarrow (1 - t)a^0 + ta^1$    # Noise action

▷ Q Update
**for** all batch elements **do**

$m \sim \mathrm{Unif}\{1, \ldots, M^{\mathrm{BTT}}\}$    # Choose number of inference steps
$a^\pi \sim \pi_\theta(\cdot \mid x, m)$    # Sample action

▷ Self-Consistency
**for** all batch elements **do**

$s_t \leftarrow s_\theta(a^t, t, h \mid x)$    # First small step
$a^{t+h} \leftarrow a^t + s_t h$    # Follow ODE
$s_{t+h} \leftarrow s_\theta(a^{t+h}, t + h, h \mid x)$    # Second small step
$s_{\mathrm{target}} \leftarrow \mathrm{stopgrad}(s_t + s_{t+h} \mid x)/2$    # Self-consistency target

▷ Flow Matching
**for** all batch elements **do**

$h \leftarrow 1/M^{\mathrm{disc}}$    # Use smallest step size
$s_{\mathrm{target}} \leftarrow a^1 - a^0$    # Flow-matching target

$\theta \leftarrow \nabla_\theta \|s_\theta(a^t, t, 2h \mid x) - s_{\mathrm{target}}\|^2 - \nabla_\theta Q_\phi(x, a^\pi)$    # Update actor

▷ Critic Update
**for** all batch elements **do**

$m \sim \mathrm{Unif}\{1, \ldots, M^{\mathrm{BTT}}\}$    # Choose number of inference steps
$a^\pi_{x'} \sim \pi_\theta(\cdot \mid x', m)$    # Sample action

$\phi \leftarrow \nabla_\phi \left(Q_\phi(x, a^1) - r - \gamma Q_\phi(x', a^\pi_{x'})\right)^2$    # Update critic

## 3   Scaling Offline Reinforcement Learning

At a high-level, offline RL aims to optimize a policy subject to some regularization constraint, which we can formulate as

$$\underset{\pi \in \Pi}{\mathrm{argmax}} \quad \underbrace{J_\mathcal{D}(\pi)}_{\texttt{Policy Optimization}} - \underbrace{\alpha R(\pi, \pi_B)}_{\texttt{Regularization}} \tag{6}$$

where $J_\mathcal{D}(\pi)$ is the expected return over offline dataset $\mathcal{D}$, $\pi_B$ is the offline data generating policy, and $R(\pi, \pi_B)$ is a regularization term (e.g. a divergence measure between $\pi$ and $\pi_B$).

The regularization constraint can be implemented through a behavioral cloning (BC)-style loss [Wu et al., 2019], such as in the behavior-regularized actor-critic formulation [Wu et al., 2019, Fujimoto and Gu, 2021, Tarasov et al., 2023a, Park et al., 2025]

$$\underset{\pi \in \Pi}{\mathrm{argmax}} \, \mathbb{E}_{x, a \sim \mathcal{D}} \left[ \mathbb{E}_{a^\pi \sim \pi_\theta(\cdot \mid x)} \left[ \underbrace{Q_\phi(x, a^\pi)}_{\texttt{Q Loss}} + \underbrace{\alpha \log \pi_\theta(a \mid x)}_{\texttt{BC Loss}} \right] \right] \tag{7}$$

where the $Q_\phi$ function is trained via minimizing the Bellman error. In order to incorporate generative models, the BC loss can be replaced with score matching or flow matching [Wang et al., 2022, Park et al., 2024a]. However, the core challenge of performing offline RL with generative models is the policy optimization component, due to the iterative nature of the noise sampling process.

### 3.1   Scalable Offline Reinforcement Learning (SORL)

**Motivation.** In order to scale offline RL, we desire an efficiently trained expressive policy that is scalable under any inference budget. Our key insight is that, by incorporating self-consistency into training, SORL can vary the number of denoising steps used during policy optimization (i.e. the

number of backpropagation through time steps), regularization to offline data (i.e. the total number of discretization steps), and inference (i.e. the number of inference steps)—thus enabling both efficient training of highly expressive policy classes *and* inference-time scaling. Unlike two-stage methods, which take existing diffusion models and later distill one-step capabilities into them, SORL is a unified model in which varying-step inference is learned by a single network in one training run.[2]

**Policy Class and Inference Procedure.** We model our policy by the shortcut function $s_\theta$, and sample actions via the Euler method with the shortcut function $s_\theta$. The full procedure is presented in Algorithm 2. Slightly overloading notation, we condition on the number of inference steps $m$ when generating actions from the policy (i.e. $a \sim \pi_\theta(\cdot \mid x, m)$). At test-time, we sample actions using $M^{\text{inf}}$ inference steps. Note that, since the inference process corresponds to solving a deterministic ODE, which is approximated using the Euler method, we can perform backpropagation through time on actions sampled via Algorithm 2 during training.

**Actor Loss.** We present SORL's full training procedure in Algorithm 1. There are three components to SORL's training: the Q update, the regularization to offline data, and the self-consistency:

$$\mathcal{L}_\pi(\theta) = \underbrace{\mathcal{L}_{\text{QL}}(\theta)}_{\text{Q Loss}} + \underbrace{\mathcal{L}_{\text{FM}}(\theta)}_{\text{Flow Matching Loss}} + \underbrace{\mathcal{L}_{\text{SC}}(\theta)}_{\text{Self-Consistency Loss}} \tag{8}$$

*(1) Q Update.* For the Q update, we first sample actions via the inference procedure in Algorithm 2, using a maximum of $M^{\text{BTT}}$ steps (i.e. backpropagating $M^{\text{BTT}}$ steps through time). The Q loss is computed with respect to this action:

$$\mathcal{L}_{\text{QL}}(\theta) = \mathbb{E}_{x \sim \mathcal{D}} \mathbb{E}_{a^\pi \sim \pi_\theta(\cdot|x)} \left[ -Q_\phi(x, a^\pi) \right] \tag{9}$$

Since $M^{\text{BTT}}$ typically is small—we experiment with $M^{\text{BTT}} = 1, 2, 4, 8$—we can backpropagate the $Q$ loss efficiently, without needing importance weighting or classifier gradients. In other words, even though sampling $a^\pi \sim \pi_\theta$ may involve multi-step generations, $\nabla_\theta \mathcal{L}_{\text{QL}}$ is still computable. Additionally, rather than use a fixed number of steps for sampling actions, we sample the number of steps uniformly from the set of powers of 2 between 1 and $M^{\text{BTT}}$.[3]

*(2) Offline Data Regularization.* We add a BC-style loss that serves as the regularization to offline data, which we implement via flow matching on the offline data, using $M^{\text{disc}}$ discretization steps. $a^0$ represents a fully noised action (i.e. noise sampled from a Gaussian), and $a^1$ represents a real action (i.e. actions sampled from the offline data $\mathcal{D}$). The flow matching loss is thus:

$$\mathcal{L}_{\text{FM}}(\theta) = \mathbb{E}_{\substack{x, a^1 \sim \mathcal{D}, a^0 \sim \mathcal{N}, \\ h \sim p(h,t)}} \left[ \left\| \underbrace{s_\theta(a^t, t, 1/M^{\text{disc}} \mid x)}_{\text{Velocity Prediction}} - \underbrace{(a^1 - a^0)}_{\text{Velocity Target}} \right\|^2 \right] \tag{10}$$

*(3) Self-Consistency.* We add a self-consistency loss to ensure that bigger jumps (e.g. the shortcut of an $m$-step procedure) are consistent with smaller jumps (e.g. the shortcut of a $2m$-step procedure):

$$\mathcal{L}_{\text{SC}}(\theta) = \mathbb{E}_{\substack{x, a^1 \sim \mathcal{D}, \\ a^0 \sim \mathcal{N}, (t,h) \sim p(t,h)}} \left[ \left\| \underbrace{s_\theta(a^t, t, 2h \mid x)}_{\text{1 Double-Step}} - \underbrace{s_{\text{target}}}_{\text{2 Single-Steps}} \right\|^2 \right] \tag{11}$$

where

$$s_{\text{target}} = \underbrace{s_\theta(a^t, t, h \mid x)/2}_{\text{1st Single-Step}} + \underbrace{s_\theta(a^{t+h}, t+h, h \mid x)/2}_{\text{2nd Single-Step}}, \tag{12}$$

and

$$\underbrace{a^{t+h}}_{\text{2nd Step's Action}} = \underbrace{a^t}_{\text{1st Step's Action}} + \underbrace{h}_{\text{Step Size}} \underbrace{s_\theta(a^t, t, h)}_{\text{Normalized Direction}} \tag{13}$$

---

[2] Note that directly regularizing to the empirical offline data, as done via flow matching in SORL, is preferable to regularizing with respect to a learned behavior cloning (BC) policy, as done by FQL, when the underlying distribution class is unknown. This is because, in the nonparametric setting, the empirical distribution is a statistically consistent estimator, and it is minimax rate-optimal under common metrics like total variation and 2-Wasserstein distance.

[3] In other words, we sample $m \sim \text{Unif}\{2^k\}_{k=0}^{\lfloor \log_2 M^{\text{BTT}} \rfloor}$.

---
**Algorithm 2:** `SORL` Action Sampling via Forward Euler Method
---
**Input:** State $x$, number of inference steps $m$
**Output:** Action $a$
$a \sim \mathcal{N}(0, I)$
$h \leftarrow 1/m$
$t \leftarrow 0$
**for** $n \in \{0, \ldots, m-1\}$ **do**
    $a \leftarrow a + h \cdot s_\theta(a, t, h \mid x)$
    $t \leftarrow t + h$
**return** $a$
---

**Critic Loss.** We train the critic via a standard Bellman error minimization, such that

$$\mathcal{L}_Q(\phi) = \left( Q_\phi(x, a^1) - r - \gamma Q_\phi^{\text{target}}(x', a_{x'}^\pi) \right)^2 \tag{14}$$

where $a_{x'}^\pi \sim \pi_\theta(\cdot \mid x')$ and $Q_\phi^{\text{target}}$ is the target network [Mnih, 2013, Park et al., 2025].

### 3.2 `SORL` Inference-Time Scaling

**The Benefit of Self-Consistency.** The loss function $\mathcal{L}_\pi(\theta)$ includes both the *flow matching error* and the *self-consistency error*. Intuitively, training on the flow matching error alone yields a flow model, from which we can sample actions by numerically solving the flow ODE in Equation 3 with the Euler method. Using more discretization steps $M^{\text{disc}}$ in the Euler method implies a smaller step size (i.e. $h_{\min} = 1/M^{\text{disc}}$ is small), leading to a smaller discretization error. However, the Euler method requires $\Theta(h_{\min}^{-1})$ number of iterations, so a smaller step size $h_{\min}$ is more computationally expensive. By incorporating the self-consistency error into the training objective, the shortcut model approximates two iteration steps of the Euler method into a single step. By iteratively approximating with larger step sizes, the shortcut model achieves discretization error comparable to that of the Euler method with a small step size, while only requiring a constant computational cost.

**Sequential Scaling.** By training a shortcut model with self-consistency, `SORL`'s policy can perform inference under varying inference budgets. In other words, `SORL` can sample actions in Algorithm 2 with a varying number of inference steps $M^{\text{inf}}$, including one step. In order to implement sequential scaling, we simply run the sampling procedure in Algorithm 2 for a greater number of steps (i.e. a larger $M^{\text{inf}}$), up to the number of discretization steps $M^{\text{disc}}$ used during training.

**Parallel Scaling.** We also desire an approach to inference-time scaling that is independent of the number of inference steps. We incorporate best-of-$N$ sampling [Fujimoto et al., 2019, Ghasemipour et al., 2021, Gui et al., 2024, Nakamoto et al., 2024, Lightman et al., 2023, Brown et al., 2024], following the simple procedure: sample actions independently and use a verifier to select the best sample. In `SORL`, we use the trained $Q$ function as the verifier. We implement best-of-$N$ sampling as follows: given a state $x$, sample $a_1, a_2, \ldots, a_N$ independently from the policy $\pi_\theta(a \mid x)$ and select the action with the largest $Q$ value, such that

$$\underset{a \in \{a_1, a_2, \ldots, a_N\}}{\arg\max} Q(x, a) \tag{15}$$

## 4 Theoretical Analysis: Regularization To Behavior Policy

Offline RL aims to learn a policy that does not deviate too far from the offline data, in order to avoid test-time distribution shift [Levine et al., 2020]. In this section, we theoretically examine the question:

> *Will `SORL` learn a policy that is regularized to the behavior of the offline data?*

Through a novel analysis of shortcut models, we prove that the is *yes*.

Table 1: **SORL's Overall Performance.** SORL achieves the best performance on 5 of the 8 environments evaluated, for a total of 40 unique tasks. The performance is averaged over 8 seeds, with standard deviations reported. The baseline results are reported from Park et al. [2025]'s extensive tuning and evaluation of baselines on OGBench tasks. We present the full results in Appendix D.

| Task Category | Gaussian | | | Diffusion | | | Flow | | | | Shortcut |
|---|---|---|---|---|---|---|---|---|---|---|---|
| | BC | IQL | ReBRAC | IDQL | SRPO | CAC | FAWAC | FBRAC | IFQL | FQL | SORL |
| OGBench antmaze-large-singletask (5 tasks) | 11 $_{\pm 1}$ | 53 $_{\pm 3}$ | 81 $_{\pm 5}$ | 21 $_{\pm 5}$ | 11 $_{\pm 4}$ | 33 $_{\pm 4}$ | 6 $_{\pm 1}$ | 60 $_{\pm 6}$ | 28 $_{\pm 5}$ | 79 $_{\pm 3}$ | **89** $_{\pm 2}$ |
| OGBench antmaze-giant-singletask (5 tasks) | 0 $_{\pm 0}$ | 4 $_{\pm 1}$ | **26** $_{\pm 8}$ | 0 $_{\pm 0}$ | 0 $_{\pm 0}$ | 0 $_{\pm 0}$ | 0 $_{\pm 0}$ | 4 $_{\pm 4}$ | 3 $_{\pm 2}$ | 9 $_{\pm 6}$ | 9 $_{\pm 6}$ |
| OGBench humanoidmaze-medium-singletask (5 tasks) | 2 $_{\pm 1}$ | 33 $_{\pm 2}$ | 22 $_{\pm 8}$ | 1 $_{\pm 0}$ | 1 $_{\pm 1}$ | 53 $_{\pm 8}$ | 19 $_{\pm 1}$ | 38 $_{\pm 5}$ | 60 $_{\pm 14}$ | 58 $_{\pm 5}$ | **64** $_{\pm 4}$ |
| OGBench humanoidmaze-large-singletask (5 tasks) | 1 $_{\pm 0}$ | 2 $_{\pm 1}$ | 2 $_{\pm 1}$ | 1 $_{\pm 0}$ | 0 $_{\pm 0}$ | 0 $_{\pm 0}$ | 0 $_{\pm 0}$ | 2 $_{\pm 0}$ | **11** $_{\pm 2}$ | 4 $_{\pm 2}$ | 5 $_{\pm 2}$ |
| OGBench antsoccer-arena-singletask (5 tasks) | 1 $_{\pm 0}$ | 8 $_{\pm 2}$ | 0 $_{\pm 0}$ | 12 $_{\pm 4}$ | 1 $_{\pm 0}$ | 2 $_{\pm 4}$ | 12 $_{\pm 0}$ | 16 $_{\pm 1}$ | 33 $_{\pm 6}$ | 60 $_{\pm 2}$ | **69** $_{\pm 2}$ |
| OGBench cube-single-singletask (5 tasks) | 5 $_{\pm 1}$ | 83 $_{\pm 3}$ | 91 $_{\pm 2}$ | **95** $_{\pm 2}$ | 80 $_{\pm 5}$ | 85 $_{\pm 9}$ | 81 $_{\pm 4}$ | 79 $_{\pm 7}$ | 79 $_{\pm 2}$ | **96** $_{\pm 1}$ | **97** $_{\pm 1}$ |
| OGBench cube-double-singletask (5 tasks) | 2 $_{\pm 1}$ | 7 $_{\pm 1}$ | 12 $_{\pm 1}$ | 15 $_{\pm 6}$ | 2 $_{\pm 1}$ | 6 $_{\pm 2}$ | 5 $_{\pm 2}$ | 15 $_{\pm 3}$ | 14 $_{\pm 3}$ | **29** $_{\pm 2}$ | 25 $_{\pm 3}$ |
| OGBench scene-singletask (5 tasks) | 5 $_{\pm 1}$ | 28 $_{\pm 1}$ | 41 $_{\pm 3}$ | 46 $_{\pm 3}$ | 20 $_{\pm 1}$ | 40 $_{\pm 7}$ | 30 $_{\pm 3}$ | 45 $_{\pm 5}$ | 30 $_{\pm 3}$ | **56** $_{\pm 2}$ | **57** $_{\pm 2}$ |

In the training objective in Equation 8, we include the BC-style flow matching loss $\mathcal{L}_{\text{FM}}$ and self-consistency loss $\mathcal{L}_{\text{SC}}$. The former ensures closeness to the offline data, while the latter allows for fast action generation. In this section, we demonstrate that this training objective can be interpreted as an instantiation of the constrained policy optimization in Equation 6. Unlike the KL-style regularization in Equation 7, the $\mathcal{L}_{\text{FM}} + \mathcal{L}_{\text{SC}}$ term in the objective function is a *Wasserstein behavioral regularization*, similar to that of Park et al. [2025]. We show that under proper conditions, the shortcut model will generate a distribution close to the target in 2-Wasserstein distance ($W_2$) in Euclidean norm. This implies that as long as we minimize $\mathcal{L}_{\text{FM}} + \mathcal{L}_{\text{SC}}$, we ensure the policies induced by the shortcut model will stay close to the behavior policy in $W_2$ distance.

Following the setup of the shortcut model in Section 2, we first assume the shortcut model $s(\cdot, \cdot, \cdot)$ is trained properly (i.e. flow-matching and the self-consistency losses are minimized well). In other words, the shortcut model for the smallest step size $s(\cdot, \cdot, \frac{1}{M})$ is close to the ground truth drift function $v_t(\cdot)$, and $s(\cdot, \cdot, 2h)$ is consistent with $s(\cdot, \cdot, h)$ on the evaluated time steps in inference.

**Assumption 1** (Small Flow Matching and Self-Consistency Losses)**.** *There exist $\epsilon_{SC} > 0$ and $\epsilon_{SC} > 0$, s.t.*

- *for all $t = 0, \frac{1}{M}, \frac{2}{M}, \ldots, 1 - \frac{1}{M}$, we have $\mathbb{E}_{z_t \sim p_t}\left[\|s(z_t, t, \frac{1}{M}) - v_t(z_t)\|_2^2\right] \leq \epsilon_{FM}^2$;*

- *for all $h = \frac{1}{M}, \frac{2}{M}, \frac{2^2}{M}, \ldots, \frac{1}{2}$, and $t = 0, h, 2h, \ldots, 1 - h$, we have*

$$\mathbb{E}_{z_t \sim p_t}\left[\left\|s(z_t, t, h)/2 + s(z'_{t+h}, t, h)/2 - s(z_t, t, 2h)\right\|_2^2\right] \leq \epsilon_{SC}^2$$

*where $z'_{t+h} = z_t + s(z_t, t, h)h$.*

**Theorem 2** (Regularization To Behavior Policy)**.** *Suppose the shortcut model $s(z, t, h)$ is $L$-Lipschitz in $z$ for all $t$ and $h$, the drift function $v_t(z)$ is $L_v$-Lipschitz in $z$ for all $t$, $\sup_t \mathbb{E}_{z_t \sim p_t}\left[\|v_t\|_2^2\right] \leq M_v$ and $L/M < 1$. If Assumption 1 holds, then for all $h = \frac{1}{M}, \frac{2}{M}, \frac{2^2}{M}, \ldots, \frac{1}{2}, 1$*

$$W_2(\hat{p}^{(h)}, p^\star) \leq \frac{1}{L} e^{\frac{3}{2}L} \left( \underbrace{\frac{eL_v}{M}(M_v + 1)}_{\text{Discretization Error}} + \underbrace{\epsilon_{FM}}_{\text{Flow Matching Error}} + \underbrace{\epsilon_{SC} \log_2 M}_{\text{Self-Consistency Error}} \right) \tag{16}$$

*where $\hat{p}^{(h)}$ is the distribution of samples generated by the shortcut model with step size $h$ and $p^\star$ is the data distribution.*[4]

**Discretization Error.** In the upper bound, the term $\frac{eL_v}{M}(M_v + 1)$ corresponds to the *Euler discretization error*—a well-known quantity in numerical ODE solvers—which vanishes as the number of discretization steps $M \to \infty$. This term captures the inherent error from approximating continuous flows with finite-step shortcut models.

---

[4]For a cleaner presentation, we consider the unconditional setting and show a uniform upper bound on the Wasserstein distance for all step size $h$. We defer an $h$-dependent upper bound to Appendix C.

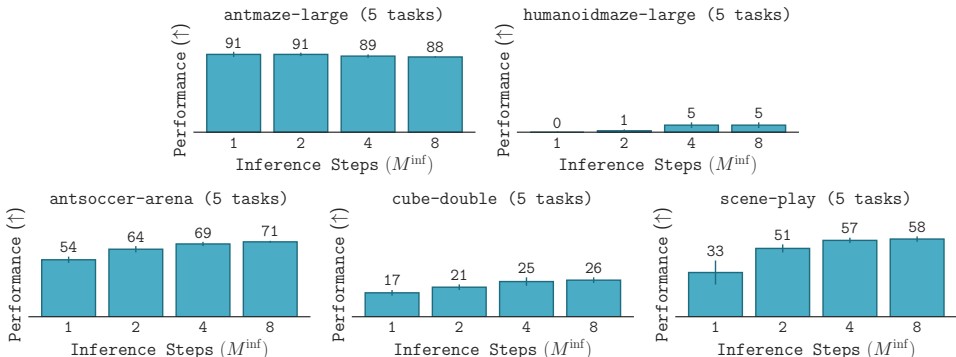

Figure 1: **SORL's Sequential Scaling.** For a fixed training budget, SORL generally improves performance with greater test-time compute. We fix a training budget of discretization steps and back-propagation steps through time ($M^{\text{disc}} = M^{\text{BTT}} = 8$) and vary the inference budget via the number of inference steps $M^{\text{inf}}$. The performance is averaged over 8 seeds for each task, with 5 tasks per environment, and standard deviations reported.

**Flow Matching and Self-Consistency Errors.** The terms $\epsilon_{\text{FM}}$ and $\epsilon_{\text{SC}} \log_2 M$ account for the training approximation errors: $\epsilon_{\text{FM}}$ measures the deviation between the shortcut models predicted velocity and the true drift under small step sizes, while $\epsilon_{\text{SC}} \log_2 M$ captures the cumulative consistency error over varying step sizes. Notably, when both $\mathcal{L}_{\text{FM}}$ and $\mathcal{L}_{\text{SC}}$ are minimized effectively during training, these errors become negligible. Ignoring the self-consistency loss, our result is comparable to the guarantees for flow models in Roy et al. [2024]. However, our analysis incorporates self-consistency and validates it across all discretization levels, which is a novel contribution.

**Regularization To Behavior Policy.** Theorem 2 shows that when trained properly, the shortcut model generates a distribution close to the target distribution in 2-Wasserstein distance for all step sizes $h$. Thus, the BC-style flow matching loss $\mathcal{L}_{\text{FM}}$ and self-consistency loss $\mathcal{L}_{\text{SC}}$ in the training objective (Equation 8) enforce regularization to behavior policy in Wasserstein distance. Consequently, SORL not only learns a performant policy, but also guarantees that the learned policy remains close to the offline data distribution across varying step sizes.

## 5 Experiments

In this section, we evaluate the overall performance of SORL against 10 baselines across 40 tasks. We then investigate SORL's sequential scaling and parallel scaling trends. Furthermore, we present additional results in Appendix D and ablation studies in Appendix E.

### 5.1 Experimental Setup

**Environments and Tasks.** We evaluate SORL on locomotion and manipulation robotics tasks in the OGBench task suite [Park et al., 2024a]. The experimental setup in this section follows the setup suggested by Park et al. [2024a, 2025]. We document the complete implementation details in Appendix F. Following Park et al. [2025], we use OGBench's reward-based `singletask` variants for all experiments, which are best suited for reward-maximizing RL.

**Baselines.** We evaluate against three Gaussian-based offline RL algorithms (BC [Pomerleau, 1988], IQL [Kostrikov et al., 2021], ReBRAC [Tarasov et al., 2023a]), three diffusion-based algorithms (IDQL [Hansen-Estruch et al., 2023], SRPO [Chen et al., 2023], CAC [Ding and Jin, 2023]), and four flow-based algorithms (FAWAC [Nair et al., 2020, Park et al., 2025], FBRAC [Zhang et al., 2025, Park et al., 2025], IFQL [Wang et al., 2022, Park et al., 2025], FQL [Park et al., 2025]). For the baselines we compare against in this paper, we report results from Park et al. [2025], who performed an extensive tuning and evaluation of the aforementioned baselines on OGBench tasks. We provide a thorough discussion of the baselines in Appendix F.

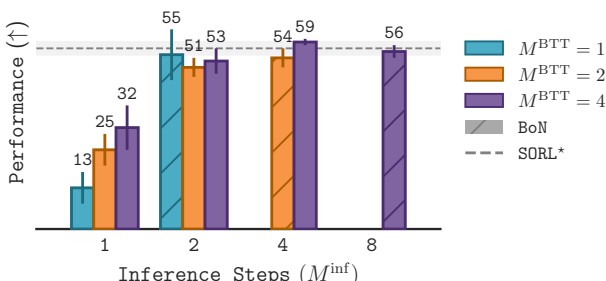

Figure 2: SORL's **Parallel Scaling.** SORL generalizes to *new* inference steps at test-time, beyond what was optimized through backpropagation during training. For each fixed training budget (i.e. fixed number of discretization steps $M^{\mathrm{disc}}$ and backpropagation through time steps $M^{\mathrm{BTT}}$), we evaluate with varying inference steps $M^{\mathrm{inf}}$. $M^{\mathrm{BTT}}$ denotes the maximum number of steps used for backpropagation through time in the $Q$ update. The / hatch denotes best-of-$N$ sampling, with $N = 8$, where the number of inference steps is *greater* than the number of backpropagation steps through time (i.e. $M^{\mathrm{inf}} > M^{\mathrm{BTT}}$). SORL$^\star$ denotes the best performance achieved by SORL in Table 1. Results are averaged over 8 seeds for each of the 5 tasks.

**Evaluation.** Following Park et al. [2025], we ensure a fair comparison by using the same network size, number of gradient steps, and discount factor for all algorithms. Furthermore, we hyperparameter tune SORL with a similar training budget to the baselines: we only tune one of SORL's training parameters on the *default* task in each environment. We average over 8 seeds per task and report standard deviations in tables. We bold values at 95% of the best performance in tables. For SORL, we use 8 discretization steps during training (2 fewer than the baselines), since SORL requires that discretization steps be powers of 2. We detail all other parameters in Appendix F.

## 5.2 Experimental Results

**Q: Does SORL perform well on different environments?**

*Yes, SORL achieves the best performance on the majority of the diverse set of 40 tasks.*

We present SORL's overall performance across a range of environments in Table 1. Notably, SORL achieves the best performance on 5 out of 8 environments, including substantial improvements over the baselines on `antmaze-large` and `antsoccer-arena`. The results suggest that while distillation approaches like FQL [Park et al., 2025] can achieve high performance, some tasks require greater expressiveness or precision than can be achieved from one-step policies.

**Q: For a fixed training budget, can SORL improve performance purely by test-time scaling?**

*Yes, SORL's performance is improved by increasing the number of inference steps at test-time.*

We investigate SORL's sequential scaling by plotting the results of varying inference steps $M^{\mathrm{inf}}$ given the same, fixed training budget (i.e. holding the discretization steps $M^{\mathrm{disc}}$ and the steps backpropagated through time $M^{\mathrm{BTT}}$ constant), for $M^{\mathrm{inf}} \leq M^{\mathrm{BTT}}$. We plot the results in Figure 1. The results show noticeable improvement in performance as the number of inference steps increases, suggesting that SORL scales positively with greater test-time compute.

**Q: Can SORL make up for less training-time compute with greater test-time compute?**

*Yes, given less training compute, SORL can match optimal performance with greater test-time compute.*

We reduce the training budget by reducing the steps of backpropagation through time from $M^{\mathrm{BTT}} = 8$ for Table 1 and Figure 1 to $M^{\mathrm{BTT}} = 1, 2, 4$ for Figure 2. However, through a combination of inference sequential and parallel scaling, we recover the performance of the optimal policy from the best training budget (SORL$^\star$).

**Q: At test-time, can** `SORL` **generalize to inference steps beyond what it was trained on?**

*Yes, through sequential and parallel scaling,* `SORL` *can use a greater number of inference steps at test-time than the number of backpropagation steps used during training.*

Given a fixed training budget (i.e. a fixed number of discretization steps $M^{\text{disc}}$ and backpropagation steps through time $M^{\text{BTT}}$), we evaluate on an increasing number of inference steps $M^{\text{inf}}$, coupled with best-of-$N$ sampling. The bar colors in Figure 2 denote different training compute, dictated by the number of backpropagation through time steps $M^{\text{BTT}}$. We apply best-of-$N$ sampling on a greater number of inference steps than were used for backpropagation through time (i.e. $M^{\text{inf}} > M^{\text{BTT}}$), thus testing `SORL`'s ability to generalize to inference steps beyond what it was backpropagated on. From Figure 2, `SORL` *can* use more inference steps than the number of backpropagation steps used during training (i.e. the number of steps used for backpropagation through time $M^{\text{BTT}}$), up to a performance saturation point (approximately $M^{\text{BTT}} = 4$).

Although Figure 2 shows that best-of-$N$ sampling can improve performance, the gain is *not* theoretically guaranteed. Our verifier is the *learned* value estimator, not a ground-truth reward, so there is no statistical benefit to learning a verifier [Swamy et al., 2025].[5] Empirically, however, best-of-$N$ effectively functions as an additional, inference-time *policy-extraction* step, searching over nearby actions and selecting the one with the highest $Q_\phi$—similar to Park et al. [2024b]'s work. Thus, the post-hoc filtering may still be empirically beneficial in regaining performance lost to imperfect optimization.

## 6    Related Work

The goal of offline reinforcement learning (RL) [Levine et al., 2020] is to learn a policy solely through previously collected data, without further interaction with the environment. The core idea of much of the prior work is to maximize returns while minimizing a discrepancy measure between the state-action distribution of the dataset and that of the learned policy. This goal has been pursued through various strategies: behavioral regularization [Nair et al., 2020, Fujimoto and Gu, 2021, Tarasov et al., 2023a], conservative value estimation [Kumar et al., 2020], in-distribution maximization [Kostrikov et al., 2021, Xu et al., 2023, Garg et al., 2023], out-of-distribution detection [Yu et al., 2020, Kidambi et al., 2020, An et al., 2021, Nikulin et al., 2023], and representing policy models using generative modeling Chen et al. [2021], Janner et al. [2021, 2022], Park et al. [2025]. Motivated by the recent success of iterative generative modeling techniques, such as denoising diffusion [Sohl-Dickstein et al., 2015, Ho et al., 2020, Song et al., 2021] and flow matching [Lipman et al., 2024, Esser et al., 2024], the use of generative models as a policy class for imitation learning and reinforcement learning has shown promise due to its expressiveness for multimodal action distributions [Wang et al., 2022, Ren et al., 2024a, Wu et al., 2024, Black et al., 2024]. However, its iterative noise sampling process is computationally inefficient [Ding and Jin, 2023]. Some methods utilize a two-stage approach: first training an iterative sampling model before then distilling it Frans et al. [2024], Ho et al. [2020], Meng et al. [2023], but this may lead to performance degradation and introduce complexity from the distillation models, along with error compounding across the two-stage procedure. Consistency models [Song and Dhariwal, 2023] are another type of unified model, but they rely on extensive bootstrapping, such as requiring a specific learning schedule during training [Frans et al., 2024], making them difficult to train.

## 7    Discussion

We introduce `SORL`: a simple, efficient one-stage training procedure for expressive policies in offline RL. `SORL`'s key property is *self-consistency*, which enables expressive inference under *any* inference budget, including one-step. Theoretically, we prove that `SORL` regularizes to the behavior policy, using a novel analysis of shortcut models. Empirically, `SORL` demonstrates the best performance across a range of diverse tasks. Additionally, `SORL` can be scaled at test-time, empirically demonstrating how greater compute at inference-time can further improve performance. An avenue for future work is to investigate how `SORL` can incorporate adaptive test-time scaling [Pan et al., 2025, Ma et al., 2025], such as selecting the number of inference steps based on the gradient of the $Q$-function.

---

[5]In other words, if the actor were already acting greedily with respect to $Q_\phi$ (i.e. the case of perfect optimization), ranking additional samples by $Q_\phi$ cannot raise the expected value.

**Acknowledgments**

NED is supported by the NSF Graduate Research Fellowship under Grant No. DGE-2139899. GKS is supported by an STTR grant. KB acknowledge: This work has been made possible in part by a gift from the Chan Zuckerberg Initiative Foundation to establish the Kempner Institute for the Study of Natural and Artificial Intelligence. WS acknowledges funding from NSF IIS-2530143, NSF IIS-2154711, NSF CAREER 2339395, DARPA LANCER: LeArning Network CybERagents, Infosys Cornell Collaboration, and Sloan Research Fellowship.

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

# A  Extended Related Work

**Offline Reinforcement Learning.**   The goal of offline reinforcement learning (RL) [Levine et al., 2020] is to learn a policy solely through previously collected data, without further interaction with the environment. A considerable amount of prior work has been developed, with the core idea being to maximize returns while minimizing a discrepancy measure between the state-action distribution of the dataset and that of the learned policy. This goal has been pursued through various strategies: behavioral regularization [Nair et al., 2020, Fujimoto and Gu, 2021, Tarasov et al., 2023a], conservative value estimation [Kumar et al., 2020], in-distribution maximization [Kostrikov et al., 2021, Xu et al., 2023, Garg et al., 2023], out-of-distribution detection [Yu et al., 2020, Kidambi et al., 2020, An et al., 2021, Nikulin et al., 2023], dual formulations of RL [Lee et al., 2021, Sikchi et al., 2023], and representing policy models using generative modeling Chen et al. [2021], Janner et al. [2021, 2022], Park et al. [2025]. After training an offline RL policy, it can be further fine-tuned with additional online rollouts, which is referred to as offline-to-online RL, for which several techniques have been proposed [Lee et al., 2021, Song et al., 2022, Nakamoto et al., 2023, Ball et al., 2023, Yu and Zhang, 2023, Ren et al., 2024b].

**Reinforcement Learning with Generative Models.**   Motivated by the recent success of iterative generative modeling techniques, such as denoising diffusion [Sohl-Dickstein et al., 2015, Ho et al., 2020, Song et al., 2021] and flow matching [Lipman et al., 2024, Esser et al., 2024], the use of generative models as a policy class for imitation learning and reinforcement learning has shown promise due to its expressiveness for multimodal action distributions [Wang et al., 2022, Ren et al., 2024a, Wu et al., 2024, Black et al., 2024]. However, its iterative noise sampling process leads to a large time consumption and memory occupancy [Ding and Jin, 2023]. Some methods utilize a two-stage approach: first training an iterative sampling model before then distilling it Frans et al. [2024], Ho et al. [2020], Meng et al. [2023], but this may lead to performance degradation and introduce complexity from the distillation models, along with error compounding across the two-stage procedure. Consistency models [Song and Dhariwal, 2023] are another type of unified model, but they rely on extensive bootstrapping, such as requiring a specific learning schedule during training [Frans et al., 2024], making them difficult to train.

**Inference-Time Scaling.**   Recent advances in large language models (LLMs) [OpenAI, 2024, DeepSeek-AI, 2025] have demonstrated the ability to increase performance at inference time with more compute through parallel scaling methods [Wang et al., 2023, Gui et al., 2024, Brown et al., 2024, Pan et al., 2025], sequential scaling methods that extend the depth of reasoning by increasing the chain of thought budget [Wei et al., 2023, Muennighoff et al., 2025, Qin et al., 2025], and self-correcting methods [Madaan et al., 2023, Qu et al., 2024, Kumar et al., 2024].

Generative models such as diffusion and flow models inherently support inference-time sequential scaling by varying the number of denoising steps [Sohl-Dickstein et al., 2015, Lipman et al., 2024], and recent work extends their inference-time scaling capabilities through parallel scaling methods via reward-guided sampling [Ma et al., 2025, Singhal et al., 2025]. However, prior reinforcement learning methods that use generative policies lose this inherent inference-time sequential scaling since they require the same number of denoising steps at both training and inference [Wang et al., 2022, Kang et al., 2023, Zhang et al., 2024]. In contrast, SORL supports both sequential and parallel scaling at inference time, allowing for dynamic trade-offs between compute and performance, and improved action selection via the learned $Q$ function.

Independent of generative models, prior work has proposed applying a form of best-of-$N$ sampling to actions from the *behavior* policy (i.e. *behavioral* candidates) [Chen et al., 2022, Fujimoto et al., 2019, Ghasemipour et al., 2021, Hansen-Estruch et al., 2023, Park et al., 2024b, Nakamoto et al., 2024]. Park et al. [2024b] proposed two methods of test-time policy improvement, by using the gradient of the $Q$-function. One of Park et al. [2024b]'s approaches relies on leveraging test-time states, which SORL's parallel scaling method does not require. The second approach proposed by Park et al. [2024b] adjusts actions using the gradient of the learned $Q$-function, which is conceptually similar to our approach of best-of-$N$ sampling with the $Q$-function verifier. However, their method requires an additional hyperparameter to tune the update magnitude in gradient space.

# B   Limitations

As noted in Section 5.2, the positive trend in parallel scaling may not occur across all environments. Our approach to parallel scaling uses the Q function as a verifier. If the learned Q function is inaccurate or out of distribution, then additional optimization may not be beneficial [Levine et al., 2020]. Additionally, while SORL is highly *flexible*, allowing for scaling of both training-time and inference-time compute budgets, SORL generally has a longer training runtime than FQL, one of the fastest flow-based baselines [Park et al., 2025], as evidenced by the runtime comparison (Figure 3) in Appendix D. We believe that trading off greater training runtime for improved performance is desirable in offline RL, since training does not require interaction with an expert, environment, or simulator. Finally, as noted in Park et al. [2025], *offline* RL algorithms, including SORL, lack a principled exploration strategy that may be necessary for attaining optimal performance in *online* RL.

## C  Proofs

We first present the upper bound on the Wasserstein distance with explicit dependency on $h$.

**Theorem 3** (Restatement of Theorem 2 With Explicit Dependency on $h$). *Suppose the shortcut model $s(z,t,h)$ is L-Lipschitz in $z$ for all $t$ and $h$, the drift function $v_t(z)$ is $L_v$-Lipschitz in $z$ for all $t$, $\sup_t \mathbb{E}_{z_t \sim p_t} \left[ \|v_t\|_2^2 \right] \leq M_v$ and $L/M < 1$. If Assumption 1 holds, then for all $h = \frac{1}{M}, \frac{2}{M}, \frac{2^2}{M}, \ldots, \frac{1}{2}, 1$*

$$W_2(\hat{p}^{(h)}, p^\star) \leq \frac{1}{L} \left( (1 + Lh)^{\frac{1}{h}} - 1 \right) \exp\left( \frac{1}{2} Lh \right) \left( \frac{eL_v}{M} (M_v + 1) + \epsilon_{FM} + \epsilon_{SC} \log_2(Mh) \right)$$

*where $\hat{p}^{(h)}$ is the distribution of samples generated by the shortcut model with step size $h$ and $p^\star$ is the data distribution.*

### C.1  Proof of Theorem 2 / 3

This theorem formalizes the key claim: minimizing the training objective keeps the learned policy close to the behavior policy. This closeness is measured using a strong metric—Wasserstein distance—implying the model will not "drift" far from the offline data distribution during generation. Under Lipschitz assumptions and small flow-matching/self-consistency loss (Assumption 1), the shortcut model generates samples whose distribution is close to the target data distribution in 2-Wasserstein distance ($W_2$), uniformly over all discretization step sizes $h$.

**Proof Sketch.**  At a high level, our goal is to show that, starting from the same $z_0 \sim \mathcal{N}(0, I)$, running the sampling process of the shortcut model and running the ground truth flow ODE yield similar output measured by square error averaged over $z_0 \sim \mathcal{N}(0, I)$.

This argument constructs a coupling with marginal distributions $\hat{p}^{(h)}$ and $p^\star$ and small transport cost. By definition, the Wasserstein distance between $\hat{p}^{(h)}$ and $p^\star$ is also small.

Our proof follows from three steps, described below.

**(1) Small Step Error.**  We first show that the shortcut model provides a good local approximation to the true dynamics when trained well at the smallest step size. In other words, under small flow-matching error and Lipschitz drift dynamics, the shortcut model has bounded error when running a single inference step with the smallest stepsize:

**Lemma 4** (Single-Step Error with Minimum Step-Size). *Let $h_0 = \frac{1}{M}$ be the smallest stepsize. If*

- *for all $t = 0, h_0, 2h_0, \ldots, 1 - h_0$, $E_{z_t \sim P_t} \left[ \|s(z_t, t, h_0) - v_t(z_t))\|_2^2 \right] \leq \epsilon_{FM}^2$,*

- *$v_t(x)$ is L-Lipschitz in both $t$ and $x$;*

- *for all $t = 0, h_0, 2h_0, \ldots, 1 - h_0$, $E_{z_t \sim P_t} \left[ \|v_t(z_t))\|_2^2 \right] \leq M_v^2$,*

*then*

$$\mathbb{E}_{z_t \sim P_t} \left[ \|z_t + s(z_t, t, h_0)h_0 - F(z_t, t, t + h_0)\|_2^2 \right] \leq h_0^2 \left( L_v e^{L_v h_0} h_0 (M_v + 1) + \epsilon_{FM} \right)^2. \quad (17)$$

**(2) Larger Step Error.**  We further demonstrate that the single inference step remains accurate even at coarser step sizes, thanks to the self-consistency constraint:

**Lemma 5** (Single-Step Error with Step Size $h$). *Let $h_0 = \frac{1}{M}$ be the smallest stepsize. If*

- *for all $h'$ and $t = 0, h', 2h', \ldots, 1 - h'$, $\mathbb{E}\left[ \|F^{(2h')}(z_t, t, t + 2h') - F^{(h')}(z_t, t, t + 2h')\|_2^2 \right] \leq 4h'^2 \epsilon_{SC}^2;$*

- *for all $t = 0, h_0, 2h_0, \ldots, 1 - h_0$, $\mathbb{E}_{z_t \sim P_t} \left[ \|z_t + s(z_t, t, h_0)h_0 - F(z_t, t, t + h_0)\|_2^2 \right] \leq h_0^2 \epsilon^2;$*

- *for all $h'$ and $t = 0, h', 2h', \ldots, 1 - h'$, $s(\cdot, t, h')$ is L-Lipschitz,*

*then for $h = 2^n h_0$, and for all $t = 0, h, \dots, 1 - h$,*

$$\sqrt{\mathbb{E}\left[\|F^{(h')}(z_t, t, t + h') - F(z_t, t, t + h')\|_2^2\right]} \le h \exp\left(\frac{1}{2}Lh\right)(\epsilon + n\epsilon_{SC}). \qquad (18)$$

**(3) Composed Error Over Multiple Steps.** Finally, even with repeated applications, the shortcut models errors remain controlled, showing stability over long horizons. In other words, if the single inference step error is small and $s$ is $L$-Lipschitz, then the multi-step trajectory error is bounded:

**Lemma 6** (Error of $1/h$-Step Inference). *If $\mathbb{E}_{z_t \sim P_t}\left[\|z_t + s(z_t, t, h)h - F(z_t, t, t + h)\|_2^2\right] \le h^2 \epsilon^2$ for all $t = 0, h, 2h, \dots, 1$, and $s(\cdot, t, h)$ is $L$-Lipschitz, then*

$$\sqrt{\mathbb{E}\left[\|\hat{z}_1^{(h)} - z_1\|_2^2\right]} \le \left((1 + Lh)^{\frac{1}{h}} - 1\right)\frac{\epsilon}{L}. \qquad (19)$$

**Summary: Aggregation of Error Bounds.** Given these insights, we finish the proof of the main theorem by applying the lemmas in order.

*Proof of Theorem 2 / 3.* Let $h_0 = \frac{1}{M}$ be the smallest step size. Suppose $h = 2^n h_0$.

By Lemma 4,

$$\mathbb{E}_{z_t \sim P_t}\left[\|z_t + s(z_t, t, h_0)h_0 - F(z_t, t, t + h_0)\|_2^2\right] \le h_0^2 \left(L_v e^{L_v h_0} h_0 (M_v + 1) + \epsilon_{FM}\right)^2. \qquad (20)$$

By Lemma 5,

$$\sqrt{\mathbb{E}\left[\|F^{(h)}(z_t, t, t + h) - F(z_t, t, t + h)\|_2^2\right]} \le h \exp\left(\frac{1}{2}Lh\right)\left(L_v e^{L_v h_0} h_0 (M_v + 1) + \epsilon_{FM} + n\epsilon_{SC}\right). \qquad (21)$$

By Lemma 6,

$$\sqrt{\mathbb{E}\left[\|\hat{z}_1^{(h)} - z_1\|_2^2\right]} \le \frac{1}{L}\left((1 + Lh)^{\frac{1}{h}} - 1\right)\exp\left(\frac{1}{2}Lh\right)\left(L_v e^{L_v h_0} h_0 (M_v + 1) + \epsilon_{FM} + n\epsilon_{SC}\right) \qquad (22)$$

$\square$

## C.2  Proof of Lemma 4

**Proof Sketch.**  The proof's strategy is:

1. Compare the true dynamics $z_t$ with linear approximation $\bar{z}_t$ using ODE analysis.
2. Bound the deviation over time.
3. Add the flow-matching loss to account for model error.

*Proof.* Consider $t \in [t', t' + h_0]$. Let

$$z_t := F(z_{t'}, t', t) \tag{23}$$

$$\bar{z}_t := z_{t'} + v_{t'}(z_{t'})(t - t'). \tag{24}$$

By definition,

$$\frac{\mathrm{d}z_t}{\mathrm{d}t} = v_t(z_t) \tag{25}$$

$$\frac{\mathrm{d}\bar{z}_t}{\mathrm{d}t} = v_{t'}(z_{t'}) \tag{26}$$

Then

$$\frac{\mathrm{d}}{\mathrm{d}t}\|z_t - \bar{z}_t\|_2^2 = 2\left\langle z_t - \bar{z}_t, \frac{\mathrm{d}}{\mathrm{d}t}z_t - \frac{\mathrm{d}}{\mathrm{d}t}\bar{z}_t\right\rangle \tag{27}$$

$$= 2\left\langle z_t - \bar{z}_t, v_t(z_t) - v_{t'}(z_{t'})\right\rangle \tag{28}$$

$$\leq 2\|z_t - \bar{z}_t\|_2\|v_t(z_t) - v_{t'}(z_{t'})\|_2 \tag{29}$$

On the other hand, by chain rule:

$$\frac{\mathrm{d}}{\mathrm{d}t}\|z_t - \bar{z}_t\|_2^2 = 2\|z_t - \bar{z}_t\|_2\frac{\mathrm{d}}{\mathrm{d}t}\|z_t - \bar{z}_t\|_2 \tag{30}$$

By Equations 27 and 30:

$$\frac{\mathrm{d}}{\mathrm{d}t}\|z_t - \bar{z}_t\|_2 \leq \|v_t(z_t) - v_{t'}(z_{t'})\|_2 \tag{31}$$

By triangle inequality:

$$\|v_t(z_t) - v_{t'}(z_{t'})\|_2 \leq \|v_t(z_t) - v_t(\bar{z}_t)\|_2 + \|v_t(\bar{z}_t) - v_t(z_{t'})\|_2 + \|v_t(z_{t'}) - v_{t'}(z_{t'})\|_2 \tag{32}$$

Because $v_t(x)$ is $L_v$-Lipschitz in $x$,

$$\|v_t(z_t) - v_t(\bar{z}_t)\|_2 \leq L_v\|z_t - \bar{z}_t\|_2. \tag{33}$$

Because $\bar{z}_t = z_{t'} + v_{t'}(z_{t'})(t - t')$ and $v_t(x)$ is $L_v$-Lipschitz in $x$,

$$\|v_t(\bar{z}_t) - v_t(z_{t'})\|_2 = \|v_t(z_{t'} + v_{t'}(z_{t'})(t - t')) - v_t(z_{t'})\|_2 \tag{34}$$

$$\leq L_v\|v_{t'}(z_{t'})\|_2(t - t') \tag{35}$$

$$\leq L_v\|v_{t'}(z_{t'})\|_2 h_0. \tag{36}$$

Because $v_t(x)$ is $L_v$-Lipschitz in $t$,

$$\|v_t(z_{t'}) - v_{t'}(z_{t'})\|_2 \leq L_v(t - t') \leq L_v h_0. \tag{37}$$

Thus

$$\frac{\mathrm{d}}{\mathrm{d}t}\|z_t - \bar{z}_t\|_2 \leq L_v\|z_t - \bar{z}_t\|_2 + L_v(\|v_{t'}(z_{t'})\|_2 + 1)h_0 \tag{38}$$

Multiplying $e^{-L_v(t-t')}$ on both side, we get:

$$e^{-L_v(t-t')}\frac{\mathrm{d}}{\mathrm{d}t}\|z_t - \bar{z}_t\|_2 \leq e^{-L_v(t-t')}L_v\|z_t - \bar{z}_t\|_2 + e^{-L_v(t-t')}L_v(\|v_{t'}(z_{t'})\|_2 + 1)h_0 \tag{39}$$

By rearranging:

$$\frac{\mathrm{d}}{\mathrm{d}t}\left(e^{-L_v(t-t')}\|z_t - \bar{z}_t\|_2\right) \leq e^{-L_v(t-t')}L_v(\|v_{t'}(z_{t'})\|_2 + 1)h_0 \tag{40}$$

$$\leq L_v(\|v_{t'}(z_{t'})\|_2 + 1)h_0. \tag{41}$$

Because $z_{t'} = \bar{z}_{t'}$, by integrating both side over $t \in [t', t'+h_0]$, we have:

$$e^{-L_v h_0}\|z_{t'+h_0} - \bar{z}_{t'+h_0}\|_2 \leq L_v(\|v_{t'}(z_{t'})\|_2 + 1)h_0^2. \tag{42}$$

Thus

$$\|z_{t'+h_0} - \bar{z}_{t'+h_0}\|_2 \leq L_v(\|v_{t'}(z_{t'})\|_2 + 1)e^{L_v h_0}h_0^2. \tag{43}$$

Taking square and expectation on both sides, we have:

$$\mathbb{E}\left[\|z_{t'+h_0} - \bar{z}_{t'+h_0}\|_2^2\right] \leq L_v^2 e^{2L_v h_0}h_0^4\mathbb{E}\left[(\|v_{t'}(z_{t'})\|_2 + 1)^2\right]. \tag{44}$$

Thus

$$\sqrt{\mathbb{E}\left[\|z_{t'+h_0} - \bar{z}_{t'+h_0}\|_2^2\right]} \leq L_v e^{L_v h_0}h_0^2\sqrt{\mathbb{E}\left[(\|v_{t'}(z_{t'})\|_2 + 1)^2\right]} \tag{45}$$

$$\leq L_v e^{L_v h_0}h_0^2\left(\sqrt{\mathbb{E}\left[\|v_{t'}(z_{t'})\|_2^2\right]} + 1\right) \tag{46}$$

$$\leq L_v e^{L_v h_0}h_0^2(M_v + 1) \tag{47}$$

By definition,

$$\|\bar{z}_{t'+h_0} - \hat{z}_{t'+h_0}\|_2 = \|z_{t'} + v_{t'}(z_{t'})h_0 - (z_{t'} + s(z_{t'}, t', h_0)h_0)\|_2 \tag{48}$$

$$= \|v_{t'}(z_{t'}) - s(z_{t'}, t', h_0)\|_2 h_0 \tag{49}$$

Taking square and expectation on both sides, we have

$$\mathbb{E}\left[\|\bar{z}_{t'+h_0} - \hat{z}_{t'+h_0}\|_2^2\right] = h_0^2\mathbb{E}\left[\|v_{t'}(z_{t'}) - s(z_{t'}, t', h_0)\|_2^2\right] \leq h_0^2\epsilon_{\mathrm{FM}}^2. \tag{50}$$

Thus

$$\sqrt{\mathbb{E}_{z_{t'}}\left[\|z_{t'+h_0} - \hat{z}_{t'+h_0}\|_2^2\right]} \leq \sqrt{\mathbb{E}_{z_{t'}}\left[\|z_{t'+h_0} - \bar{z}_{t'+h_0}\|_2^2\right]} + \sqrt{\mathbb{E}_{z_{t'}}\left[\|\bar{z}_{t'+h_0} - \hat{z}_{t'+h_0}\|_2^2\right]} \tag{51}$$

$$\leq Le^{L_v h_0}h_0^2(M_v + 1) + h_0\epsilon_{\mathrm{FM}} \tag{52}$$

$$\leq h_0\left(Le^{L_v h_0}h_0(M_v + 1) + \epsilon_{\mathrm{FM}}\right) \tag{53}$$

$$\square$$

## C.3 Proof of Lemma 5

**Proof Sketch.** The proof's strategy is:

1. Define a recursive error relationship for step sizes $2^k h_0$.
2. Use inductive bounding and logarithmic scaling of error growth.
3. Solve the recursion to show controlled error amplification.

*Proof.* We define $\Delta_{h'}$ to be the maximum 1-step error induced by shortcut model with step size $h'$. Formally, for all $h' > 0$ s.t. $1/h' \in \mathbb{Z}$, we define,

$$\Delta_{h'} := \max_{t \in \{0, h' \ldots, 1-h'\}} \sqrt{\mathbb{E}\left[\|F^{(h')}(z_t, t, t+h') - F(z_t, t, t+h')\|_2^2\right]} \tag{54}$$

For all $h'$ and $t \in \{0, 2h' \ldots, 1-2h'\}$,

$$\sqrt{\mathbb{E}\left[\|F^{(2h')}(z_t, t, t+2h') - F(z_t, t, t+2h')\|_2^2\right]}$$

$$\leq \sqrt{\mathbb{E}\left[\|F^{(2h')}(z_t, t, t+2h') - F^{(h')}(z_t, t, t+2h')\|_2^2\right]}$$

$$+ \sqrt{\mathbb{E}\left[\|F^{(h')}(z_t, t, t+2h') - F(z_t, t, t+2h')\|_2^2\right]} \tag{55}$$

By assumption, the first term is bounded by $2h'\epsilon_{\text{SC}}$. We now analyze the second term. Let $\hat{z}_{t+h'} := F^{(h')}(z_t, t, t+h')$, then

$$\sqrt{\mathbb{E}\left[\|F^{(h')}(z_t, t, t+2h') - F(z_t, t, t+2h')\|_2^2\right]} \tag{56}$$

$$= \sqrt{\mathbb{E}\left[\|F^{(h')}(\hat{z}_{t+h'}, t+h', t+2h') - F(z_{t+h'}, t+h', t+2h')\|_2^2\right]} \tag{57}$$

$$\leq \sqrt{\mathbb{E}\left[\|F^{(h')}(\hat{z}_{t+h'}, t+h', t+2h') - F^{(h')}(z_{t+h'}, t+h', t+2h')\|_2^2\right]} \tag{58}$$

$$+ \sqrt{\mathbb{E}\left[\|F^{(h')}(z_{t+h'}, t+h', t+2h') - F(z_{t+h'}, t+h', t+2h')\|_2^2\right]}. \tag{59}$$

By triangle inequality,

$$\|F^{(h')}(\hat{z}_{t+h'}, t+h', t+2h') - F^{(h')}(z_{t+h'}, t+h', t+2h')\|_2 \tag{60}$$

$$= \|(\hat{z}_{t+h'} + s(\hat{z}_{t+h'}, t+h', t+2h')h') - (z_{t+h'} + s(z_{t+h'}, t+h', t+2h')h')\|_2 \tag{61}$$

$$\leq \|\hat{z}_{t+h'} - z_{t+h'}\|_2 + h' \|s(\hat{z}_{t+h'}, t+h', h') - s(z_{t+h'}, t+h', h')\|_2 \tag{62}$$

Because $s(\cdot, t, h')$ is $L$-Lipschitz (Assumption 1),

$$\|F^{(h')}(\hat{z}_{t+h'}, t+h', t+2h') - F^{(h')}(z_{t+h'}, t+h', t+2h')\|_2 \leq (1+Lh')\|\hat{z}_{t+h'} - z_{t+h'}\|_2 \tag{63}$$

Thus

$$\sqrt{\mathbb{E}\left[\|F^{(h')}(\hat{z}_{t+h'}, t+h', t+2h') - F^{(h')}(z_{t+h'}, t+h', t+2h')\|_2^2\right]} \tag{64}$$

$$\leq (1+Lh')\sqrt{\mathbb{E}\left[\|\hat{z}_{t+h'} - z_{t+h'}\|_2^2\right]} \tag{65}$$

$$= (1+Lh')\sqrt{\mathbb{E}\left[\|F^{(h')}(z_t, t, t+h') - F(z_t, t, t+h')\|_2^2\right]} \leq (1+Lh')\Delta_{h'}. \tag{66}$$

By Equation 54,

$$\sqrt{\mathbb{E}\left[\|F^{(h')}(z_{t+h'}, t+h', t+2h') - F(z_{t+h'}, t+h', t+2h')\|_2^2\right]} \leq \Delta_{h'}. \tag{67}$$

Combine everything together,

$$\sqrt{\mathbb{E}\left[\|F^{(2h')}(z_t, t, t+2h') - F(z_t, t, t+2h')\|_2^2\right]}$$

$$\leq 2h'\epsilon_{\text{SC}} + (1+Lh')\Delta_{h'} + \Delta_{h'} = 2h'\epsilon_{\text{SC}} + (2+Lh')\Delta_{h'}. \tag{69}$$

Because $t'$ is chosen arbitrarily, we have

$$\Delta_{2h'} \leq 2h'\epsilon_{\text{SC}} + (2 + Lh')\Delta_{h'}. \tag{70}$$

Let $h' := 2^k h_0$ and $A_k := \Delta_{2^k h_0}$, we have:

$$A_{k+1} \leq (2 + Lh_0 2^k)A_k + 2\epsilon_{\text{SC}}h_0 2^k. \tag{71}$$

Solving this recursion, we have:

$$A_n \leq A_0 \prod_{j=0}^{n-1}(2 + Lh_0 2^j) + 2\epsilon_{\text{SC}}h_0 \sum_{k=0}^{n-1} 2^k \prod_{j=k+1}^{n-1}(2 + Lh_0 2^j) \tag{72}$$

$$= A_0 2^n \prod_{j=0}^{n-1}\left(1 + \frac{1}{2}Lh_0 2^j\right) + 2\epsilon_{\text{SC}}h_0 \sum_{k=0}^{n-1} 2^k 2^{n-k-1} \prod_{j=k+1}^{n-1}\left(1 + \frac{1}{2}Lh_0 2^j\right) \tag{73}$$

$$= A_0 2^n \prod_{j=0}^{n-1}\left(1 + \frac{1}{2}Lh_0 2^j\right) + 2^n \epsilon_{\text{SC}}h_0 \sum_{k=0}^{n-1} \prod_{j=k+1}^{n-1}\left(1 + \frac{1}{2}Lh_0 2^j\right). \tag{74}$$

Because $\ln(1 + x) \leq x$ for all $x > -1$, we have

$$A_n \leq A_0 2^n \exp\left(\sum_{j=0}^{n-1} \ln\left(1 + \frac{1}{2}Lh_0 2^j\right)\right) \tag{75}$$

$$+ 2^n \epsilon_{\text{SC}}h_0 \sum_{k=0}^{n-1} \exp\left(\sum_{j=k+1}^{n-1} \ln\left(1 + \frac{1}{2}Lh_0 2^j\right)\right) \tag{76}$$

$$\leq A_0 2^n \exp\left(\frac{1}{2}Lh_0 \sum_{j=0}^{n-1} 2^j\right) + 2^n \epsilon_{\text{SC}}h_0 \sum_{k=0}^{n-1} \exp\left(\frac{1}{2}Lh_0 \sum_{j=k+1}^{n-1} 2^j\right) \tag{77}$$

$$= A_0 2^n \exp\left(\frac{1}{2}Lh_0(2^n - 1)\right) + 2^n \epsilon_{\text{SC}}h_0 \sum_{k=0}^{n-1} \exp\left(\frac{1}{2}Lh_0 2^{k+1}(2^{n-k-1} - 1)\right) \tag{78}$$

$$\leq A_0 2^n \exp\left(\frac{1}{2}Lh_0 2^n\right) + 2^n \epsilon_{\text{SC}}h_0 \sum_{k=0}^{n-1} \exp\left(\frac{1}{2}Lh_0 2^n\right) \tag{79}$$

$$= 2^n h_0 \exp\left(\frac{1}{2}Lh_0 2^n\right)\left(\frac{A_0}{h_0} + n\epsilon_{\text{SC}}\right) \tag{80}$$

$$= h \exp\left(\frac{1}{2}Lh\right)\left(\frac{A_0}{h_0} + n\epsilon_{\text{SC}}\right) \tag{81}$$

Because $\mathbb{E}_{z_t \sim P_t}\left[\|z_t + s(z_t, t, h_0)h_0 - F(z_t, t, t + h_0)\|_2^2\right] \leq h_0^2 \epsilon^2$, we have $A_0 \leq h_0 \epsilon$. Thus

$$\Delta_h = \Delta_{2^n h_0} = A_n \leq h \exp\left(\frac{1}{2}Lh\right)(\epsilon + n\epsilon_{\text{SC}}). \tag{82}$$

$\square$

## C.4 Proof of Lemma 6

**Proof Sketch.** The proof's strategy is:

1. Define the recurrence over time steps for error accumulation.
2. Use Lipschitz continuity to control error propagation.
3. Solve the recurrence analytically.

*Proof.*

$$\sqrt{\mathbb{E}\left[\|\hat{z}_{t+h}^{(h)} - z_{t+h}\|_2^2\right]} \leq \sqrt{\mathbb{E}\left[\|\hat{z}_t^{(h)} + s(\hat{z}_t^{(h)}, t, h)h - (z_t + s(z_t, t, h)h)\|_2^2\right]}$$

$$+ \sqrt{\mathbb{E}\left[\|(z_t + s(z_t, t, h)h) - z_{t+h}\|_2^2\right]} \tag{83}$$

$$\leq (1 + Lh)\sqrt{\mathbb{E}\left[\|\hat{z}_t^{(h)} - z_t\|_2^2\right]} + h\epsilon \tag{84}$$

Because $\mathbb{E}[\|\hat{z}_0^{(h)} - z_0\|_2^2] = 0$, by solving the recursion,

$$\sqrt{\mathbb{E}\left[\|\hat{z}_1^{(h)} - z_1\|_2^2\right]} \leq \left((1 + Lh)^{\frac{1}{h}} - 1\right)\frac{\epsilon}{L} \tag{85}$$

$\square$

# D  Full Results

Table 2:  **SORL's Overall Performance By Task.**  We present the full results on the OGBench tasks. (*) indicates the default task in each environment. The results are averaged over 8 seeds with standard deviations reported. The baseline results are reported from Park et al. [2025]'s extensive tuning and evaluation of baselines on OGBench tasks.

| | Gaussian | | | Diffusion | | | Flow | | | | Shortcut |
|---|---|---|---|---|---|---|---|---|---|---|---|
| Task | BC | IQL | ReBRAC | IDQL | SRPO | CAC | FAWAC | FBRAC | IFQL | FQL | SORL |
| antmaze-large-navigate-singletask-task1-v0 (*) | $0_{\pm 0}$ | $48_{\pm 9}$ | $\mathbf{91}_{\pm 10}$ | $0_{\pm 0}$ | $0_{\pm 0}$ | $42_{\pm 7}$ | $1_{\pm 1}$ | $70_{\pm 20}$ | $24_{\pm 17}$ | $80_{\pm 8}$ | $\mathbf{93}_{\pm 2}$ |
| antmaze-large-navigate-singletask-task2-v0 | $6_{\pm 3}$ | $42_{\pm 6}$ | $\mathbf{88}_{\pm 4}$ | $14_{\pm 8}$ | $4_{\pm 4}$ | $1_{\pm 1}$ | $0_{\pm 1}$ | $35_{\pm 12}$ | $8_{\pm 3}$ | $57_{\pm 10}$ | $79_{\pm 5}$ |
| antmaze-large-navigate-singletask-task3-v0 | $29_{\pm 5}$ | $72_{\pm 7}$ | $51_{\pm 18}$ | $26_{\pm 8}$ | $3_{\pm 2}$ | $49_{\pm 10}$ | $12_{\pm 4}$ | $83_{\pm 15}$ | $52_{\pm 17}$ | $\mathbf{93}_{\pm 3}$ | $\mathbf{88}_{\pm 10}$ |
| antmaze-large-navigate-singletask-task4-v0 | $8_{\pm 3}$ | $51_{\pm 9}$ | $84_{\pm 7}$ | $62_{\pm 25}$ | $45_{\pm 19}$ | $17_{\pm 6}$ | $10_{\pm 3}$ | $37_{\pm 18}$ | $18_{\pm 8}$ | $80_{\pm 4}$ | $\mathbf{91}_{\pm 2}$ |
| antmaze-large-navigate-singletask-task5-v0 | $10_{\pm 3}$ | $54_{\pm 22}$ | $\mathbf{90}_{\pm 2}$ | $2_{\pm 2}$ | $1_{\pm 1}$ | $55_{\pm 6}$ | $9_{\pm 5}$ | $76_{\pm 8}$ | $38_{\pm 18}$ | $83_{\pm 4}$ | $\mathbf{95}_{\pm 0}$ |
| antmaze-giant-navigate-singletask-task1-v0 (*) | $0_{\pm 0}$ | $0_{\pm 0}$ | $\mathbf{27}_{\pm 22}$ | $0_{\pm 0}$ | $0_{\pm 0}$ | $0_{\pm 0}$ | $0_{\pm 0}$ | $0_{\pm 1}$ | $0_{\pm 0}$ | $4_{\pm 5}$ | $12_{\pm 6}$ |
| antmaze-giant-navigate-singletask-task2-v0 | $0_{\pm 0}$ | $1_{\pm 1}$ | $\mathbf{16}_{\pm 17}$ | $0_{\pm 0}$ | $0_{\pm 0}$ | $0_{\pm 0}$ | $0_{\pm 0}$ | $4_{\pm 7}$ | $0_{\pm 0}$ | $9_{\pm 7}$ | $0_{\pm 0}$ |
| antmaze-giant-navigate-singletask-task3-v0 | $0_{\pm 0}$ | $0_{\pm 0}$ | $\mathbf{34}_{\pm 22}$ | $0_{\pm 0}$ | $0_{\pm 0}$ | $0_{\pm 0}$ | $0_{\pm 0}$ | $0_{\pm 0}$ | $0_{\pm 0}$ | $0_{\pm 1}$ | $0_{\pm 0}$ |
| antmaze-giant-navigate-singletask-task4-v0 | $0_{\pm 0}$ | $0_{\pm 0}$ | $5_{\pm 12}$ | $0_{\pm 0}$ | $0_{\pm 0}$ | $0_{\pm 0}$ | $0_{\pm 0}$ | $9_{\pm 4}$ | $0_{\pm 0}$ | $14_{\pm 23}$ | $\mathbf{25}_{\pm 18}$ |
| antmaze-giant-navigate-singletask-task5-v0 | $1_{\pm 1}$ | $19_{\pm 7}$ | $\mathbf{49}_{\pm 22}$ | $0_{\pm 1}$ | $0_{\pm 0}$ | $0_{\pm 0}$ | $0_{\pm 0}$ | $6_{\pm 10}$ | $13_{\pm 9}$ | $16_{\pm 28}$ | $6_{\pm 15}$ |
| humanoidmaze-medium-navigate-singletask-task1-v0 (*) | $1_{\pm 0}$ | $32_{\pm 7}$ | $16_{\pm 9}$ | $1_{\pm 1}$ | $0_{\pm 0}$ | $38_{\pm 19}$ | $6_{\pm 2}$ | $25_{\pm 8}$ | $\mathbf{69}_{\pm 19}$ | $19_{\pm 12}$ | $\mathbf{67}_{\pm 4}$ |
| humanoidmaze-medium-navigate-singletask-task2-v0 | $1_{\pm 0}$ | $41_{\pm 9}$ | $18_{\pm 16}$ | $1_{\pm 1}$ | $1_{\pm 1}$ | $47_{\pm 35}$ | $40_{\pm 2}$ | $76_{\pm 10}$ | $85_{\pm 11}$ | $\mathbf{94}_{\pm 3}$ | $89_{\pm 3}$ |
| humanoidmaze-medium-navigate-singletask-task3-v0 | $6_{\pm 2}$ | $25_{\pm 5}$ | $36_{\pm 13}$ | $0_{\pm 1}$ | $2_{\pm 1}$ | $\mathbf{83}_{\pm 18}$ | $19_{\pm 2}$ | $27_{\pm 11}$ | $49_{\pm 49}$ | $74_{\pm 18}$ | $83_{\pm 4}$ |
| humanoidmaze-medium-navigate-singletask-task4-v0 | $0_{\pm 0}$ | $0_{\pm 1}$ | $15_{\pm 16}$ | $1_{\pm 1}$ | $1_{\pm 1}$ | $5_{\pm 4}$ | $1_{\pm 1}$ | $1_{\pm 2}$ | $1_{\pm 1}$ | $3_{\pm 4}$ | $1_{\pm 0}$ |
| humanoidmaze-medium-navigate-singletask-task5-v0 | $2_{\pm 1}$ | $66_{\pm 4}$ | $24_{\pm 20}$ | $1_{\pm 1}$ | $3_{\pm 3}$ | $91_{\pm 5}$ | $31_{\pm 7}$ | $63_{\pm 9}$ | $\mathbf{98}_{\pm 2}$ | $97_{\pm 2}$ | $81_{\pm 20}$ |
| humanoidmaze-large-navigate-singletask-task1-v0 (*) | $0_{\pm 0}$ | $3_{\pm 1}$ | $2_{\pm 1}$ | $0_{\pm 0}$ | $0_{\pm 0}$ | $0_{\pm 0}$ | $0_{\pm 0}$ | $0_{\pm 0}$ | $6_{\pm 2}$ | $7_{\pm 6}$ | $\mathbf{20}_{\pm 9}$ |
| humanoidmaze-large-navigate-singletask-task2-v0 | $\mathbf{0}_{\pm 0}$ | $\mathbf{0}_{\pm 0}$ | $\mathbf{0}_{\pm 0}$ | $\mathbf{0}_{\pm 0}$ | $\mathbf{0}_{\pm 0}$ | $\mathbf{0}_{\pm 0}$ | $\mathbf{0}_{\pm 0}$ | $\mathbf{0}_{\pm 0}$ | $\mathbf{0}_{\pm 0}$ | $\mathbf{0}_{\pm 0}$ | $\mathbf{0}_{\pm 0}$ |
| humanoidmaze-large-navigate-singletask-task3-v0 | $1_{\pm 1}$ | $7_{\pm 3}$ | $8_{\pm 4}$ | $3_{\pm 1}$ | $1_{\pm 1}$ | $2_{\pm 3}$ | $1_{\pm 1}$ | $10_{\pm 2}$ | $\mathbf{48}_{\pm 10}$ | $11_{\pm 7}$ | $5_{\pm 2}$ |
| humanoidmaze-large-navigate-singletask-task4-v0 | $1_{\pm 0}$ | $1_{\pm 0}$ | $1_{\pm 1}$ | $0_{\pm 0}$ | $0_{\pm 0}$ | $0_{\pm 1}$ | $0_{\pm 0}$ | $0_{\pm 0}$ | $1_{\pm 1}$ | $\mathbf{2}_{\pm 3}$ | $0_{\pm 0}$ |
| humanoidmaze-large-navigate-singletask-task5-v0 | $0_{\pm 1}$ | $1_{\pm 1}$ | $\mathbf{2}_{\pm 2}$ | $0_{\pm 0}$ | $0_{\pm 0}$ | $0_{\pm 0}$ | $0_{\pm 0}$ | $1_{\pm 1}$ | $0_{\pm 0}$ | $1_{\pm 3}$ | $0_{\pm 0}$ |
| antsoccer-arena-navigate-singletask-task1-v0 | $2_{\pm 1}$ | $14_{\pm 5}$ | $0_{\pm 0}$ | $44_{\pm 12}$ | $2_{\pm 1}$ | $1_{\pm 3}$ | $22_{\pm 2}$ | $17_{\pm 3}$ | $61_{\pm 25}$ | $77_{\pm 4}$ | $\mathbf{93}_{\pm 4}$ |
| antsoccer-arena-navigate-singletask-task2-v0 | $2_{\pm 2}$ | $17_{\pm 7}$ | $0_{\pm 1}$ | $15_{\pm 12}$ | $3_{\pm 1}$ | $0_{\pm 0}$ | $8_{\pm 1}$ | $8_{\pm 2}$ | $75_{\pm 3}$ | $88_{\pm 3}$ | $\mathbf{96}_{\pm 2}$ |
| antsoccer-arena-navigate-singletask-task3-v0 | $0_{\pm 0}$ | $6_{\pm 4}$ | $0_{\pm 0}$ | $0_{\pm 0}$ | $0_{\pm 0}$ | $8_{\pm 19}$ | $11_{\pm 5}$ | $16_{\pm 3}$ | $14_{\pm 22}$ | $\mathbf{61}_{\pm 6}$ | $55_{\pm 6}$ |
| antsoccer-arena-navigate-singletask-task4-v0 (*) | $1_{\pm 0}$ | $3_{\pm 2}$ | $0_{\pm 0}$ | $0_{\pm 1}$ | $0_{\pm 0}$ | $0_{\pm 0}$ | $12_{\pm 3}$ | $24_{\pm 4}$ | $16_{\pm 9}$ | $39_{\pm 6}$ | $\mathbf{54}_{\pm 5}$ |
| antsoccer-arena-navigate-singletask-task5-v0 | $0_{\pm 0}$ | $2_{\pm 2}$ | $0_{\pm 0}$ | $0_{\pm 0}$ | $0_{\pm 0}$ | $0_{\pm 0}$ | $9_{\pm 2}$ | $15_{\pm 4}$ | $0_{\pm 1}$ | $36_{\pm 9}$ | $\mathbf{47}_{\pm 9}$ |
| cube-single-play-singletask-task1-v0 | $10_{\pm 5}$ | $88_{\pm 3}$ | $89_{\pm 5}$ | $\mathbf{95}_{\pm 2}$ | $89_{\pm 7}$ | $77_{\pm 28}$ | $81_{\pm 9}$ | $73_{\pm 33}$ | $79_{\pm 4}$ | $\mathbf{97}_{\pm 2}$ | $\mathbf{97}_{\pm 2}$ |
| cube-single-play-singletask-task2-v0 (*) | $3_{\pm 1}$ | $85_{\pm 8}$ | $92_{\pm 4}$ | $\mathbf{96}_{\pm 2}$ | $82_{\pm 16}$ | $80_{\pm 30}$ | $81_{\pm 9}$ | $83_{\pm 13}$ | $73_{\pm 3}$ | $\mathbf{97}_{\pm 2}$ | $\mathbf{99}_{\pm 0}$ |
| cube-single-play-singletask-task3-v0 | $9_{\pm 3}$ | $91_{\pm 5}$ | $93_{\pm 3}$ | $\mathbf{99}_{\pm 1}$ | $96_{\pm 2}$ | $98_{\pm 1}$ | $87_{\pm 4}$ | $82_{\pm 12}$ | $88_{\pm 4}$ | $98_{\pm 2}$ | $\mathbf{99}_{\pm 1}$ |
| cube-single-play-singletask-task4-v0 | $2_{\pm 1}$ | $73_{\pm 6}$ | $\mathbf{92}_{\pm 3}$ | $93_{\pm 4}$ | $70_{\pm 18}$ | $91_{\pm 2}$ | $79_{\pm 6}$ | $79_{\pm 20}$ | $79_{\pm 6}$ | $94_{\pm 3}$ | $\mathbf{95}_{\pm 2}$ |
| cube-single-play-singletask-task5-v0 | $3_{\pm 3}$ | $78_{\pm 9}$ | $87_{\pm 8}$ | $90_{\pm 6}$ | $61_{\pm 12}$ | $80_{\pm 20}$ | $78_{\pm 10}$ | $76_{\pm 33}$ | $77_{\pm 7}$ | $\mathbf{93}_{\pm 3}$ | $\mathbf{93}_{\pm 3}$ |
| cube-double-play-singletask-task1-v0 | $8_{\pm 3}$ | $27_{\pm 5}$ | $45_{\pm 6}$ | $39_{\pm 19}$ | $7_{\pm 6}$ | $21_{\pm 8}$ | $21_{\pm 7}$ | $47_{\pm 11}$ | $35_{\pm 9}$ | $61_{\pm 9}$ | $\mathbf{77}_{\pm 11}$ |
| cube-double-play-singletask-task2-v0 (*) | $0_{\pm 0}$ | $1_{\pm 1}$ | $7_{\pm 3}$ | $16_{\pm 10}$ | $0_{\pm 0}$ | $2_{\pm 2}$ | $2_{\pm 1}$ | $22_{\pm 12}$ | $9_{\pm 5}$ | $\mathbf{36}_{\pm 6}$ | $33_{\pm 8}$ |
| cube-double-play-singletask-task3-v0 | $0_{\pm 0}$ | $0_{\pm 0}$ | $4_{\pm 1}$ | $17_{\pm 8}$ | $0_{\pm 1}$ | $3_{\pm 1}$ | $1_{\pm 1}$ | $4_{\pm 2}$ | $8_{\pm 5}$ | $\mathbf{22}_{\pm 5}$ | $12_{\pm 6}$ |
| cube-double-play-singletask-task4-v0 | $0_{\pm 0}$ | $0_{\pm 0}$ | $1_{\pm 1}$ | $0_{\pm 1}$ | $0_{\pm 0}$ | $0_{\pm 1}$ | $0_{\pm 0}$ | $0_{\pm 1}$ | $1_{\pm 1}$ | $5_{\pm 2}$ | $\mathbf{7}_{\pm 4}$ |
| cube-double-play-singletask-task5-v0 | $0_{\pm 0}$ | $4_{\pm 3}$ | $4_{\pm 2}$ | $1_{\pm 1}$ | $0_{\pm 0}$ | $3_{\pm 2}$ | $2_{\pm 1}$ | $2_{\pm 2}$ | $17_{\pm 6}$ | $\mathbf{19}_{\pm 10}$ | $1_{\pm 1}$ |
| scene-play-singletask-task1-v0 | $19_{\pm 6}$ | $94_{\pm 3}$ | $\mathbf{95}_{\pm 2}$ | $\mathbf{100}_{\pm 0}$ | $94_{\pm 4}$ | $\mathbf{100}_{\pm 1}$ | $87_{\pm 8}$ | $96_{\pm 8}$ | $98_{\pm 3}$ | $\mathbf{100}_{\pm 0}$ | $99_{\pm 1}$ |
| scene-play-singletask-task2-v0 (*) | $1_{\pm 1}$ | $12_{\pm 3}$ | $50_{\pm 13}$ | $33_{\pm 14}$ | $2_{\pm 2}$ | $50_{\pm 40}$ | $18_{\pm 8}$ | $46_{\pm 10}$ | $0_{\pm 0}$ | $76_{\pm 9}$ | $\mathbf{89}_{\pm 9}$ |
| scene-play-singletask-task3-v0 | $1_{\pm 1}$ | $32_{\pm 7}$ | $55_{\pm 16}$ | $94_{\pm 4}$ | $4_{\pm 4}$ | $49_{\pm 16}$ | $38_{\pm 9}$ | $78_{\pm 14}$ | $54_{\pm 19}$ | $\mathbf{98}_{\pm 1}$ | $97_{\pm 1}$ |
| scene-play-singletask-task4-v0 | $2_{\pm 2}$ | $0_{\pm 1}$ | $3_{\pm 3}$ | $4_{\pm 3}$ | $0_{\pm 0}$ | $0_{\pm 0}$ | $\mathbf{6}_{\pm 1}$ | $4_{\pm 4}$ | $0_{\pm 0}$ | $5_{\pm 1}$ | $1_{\pm 1}$ |
| scene-play-singletask-task5-v0 | $\mathbf{0}_{\pm 0}$ | $\mathbf{0}_{\pm 0}$ | $\mathbf{0}_{\pm 0}$ | $\mathbf{0}_{\pm 0}$ | $\mathbf{0}_{\pm 0}$ | $\mathbf{0}_{\pm 0}$ | $\mathbf{0}_{\pm 0}$ | $\mathbf{0}_{\pm 0}$ | $\mathbf{0}_{\pm 0}$ | $\mathbf{0}_{\pm 0}$ | $\mathbf{0}_{\pm 0}$ |

## D.1  Per-Task Results

**Q: What is SORL's overall performance on each task?**

*SORL achieves the best performance on the majority of the diverse set of tasks considered.*

We present SORL's overall performance for each individual task in Table 2. Note that the only SORL training parameter that changes between environments is the Q-loss coefficient, which is common in offline RL [Tarasov et al., 2023b, Park et al., 2024b, 2025]. The Q-loss coefficient was tuned on the default task in each environment, and the same coefficient was used on all 5 tasks in each environment. The experimental setup was kept consistent to ensure a fair comparison between SORL and the baselines reported in Park et al. [2025].

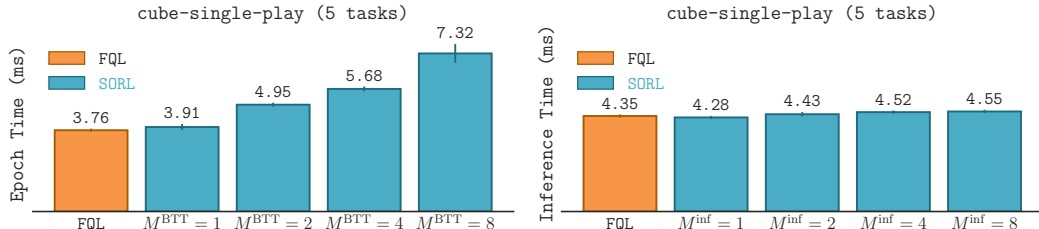

Figure 3: **Runtime Comparison.** We vary SORL's *training-time compute budget* (i.e. the number of backpropagation steps through time $M^{\text{BTT}}$) on the *left* and SORL's *inference-time compute budget* (i.e. the number of inference steps $M^{\text{inf}}$) on the *right*. The performance is averaged over 5 seeds for each task, with 5 tasks per environment, and standard deviations reported.

## D.2 Runtime Comparison

**Q: How does SORL's runtime compare to the fastest flow-based baseline, FQL?**

*SORL's scalability enables training runtime to match that of FQL under low compute budgets, while exceeding it when larger budgets are allocated. During inference, SORL maintains runtime comparable to FQL.*

We present a runtime comparison between SORL and FQL. FQL is one of the fastest flow-based baselines we consider, for both training and inference [Park et al., 2025], as FQL's policy is a one-step distillation model. We select cube-single-play because SORL and FQL achieve similar performance on the environment (Table 1). The experiments were performed on a Nvidia RTX 3090 GPU. The epoch and inference times are averaged over the first three evaluations (i.e. the first training/evaluation, training/evaluation at 100,000 gradient steps, and training/evaluation at 200,000 gradient steps). For the inference-time plot, we use the maximum training-compute budget (i.e. $M^{\text{BTT}} = 8$).

Figure 3 demonstrates SORL's scalability during both training and inference. At training time, SORL's training runtime can be shortened by decreasing the number of backpropagation steps through time. At inference time, SORL maintains similar inference times compared to FQL under varying inference-time compute budgets.

# E    Ablation Studies

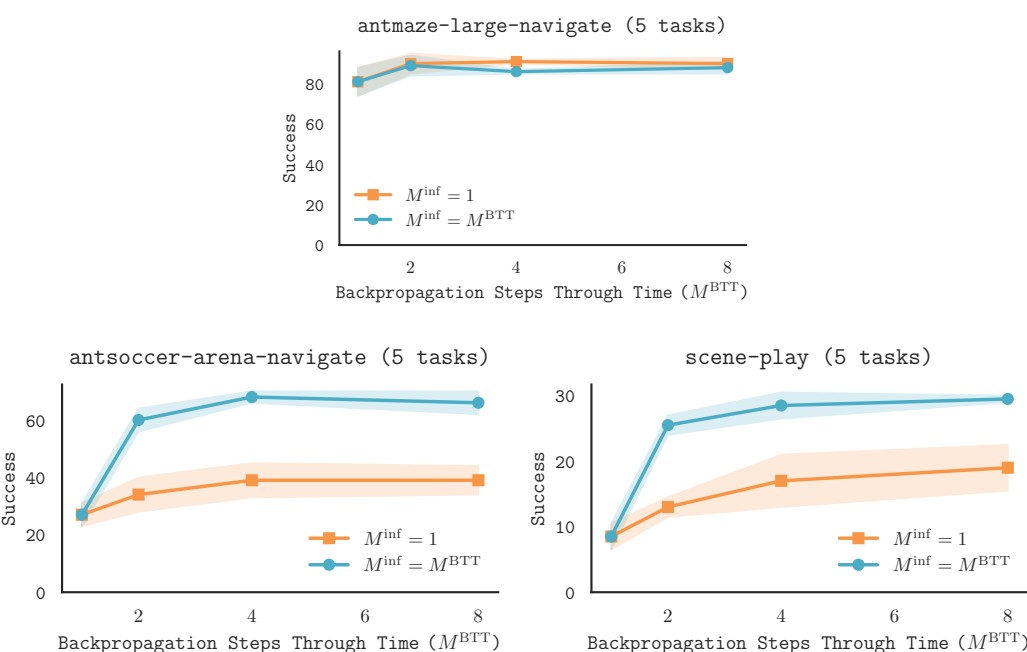

Figure 4: **Ablation Over Backpropagation Steps Through Time, $M^{\textbf{BTT}}$.** We investigate the effect of varying the training-time compute budget (i.e. the number of backpropagation steps through time $M^{\text{BTT}}$). The performance is averaged over 8 seeds for each task, with 5 tasks per environment, and standard deviations reported. We report results using one inference step (i.e. $M^{\text{inf}} = 1$) and using the same number of inference steps as backpropagation steps through time (i.e. $M^{\text{inf}} = M^{\text{BTT}}$).

## E.1    Backpropagation Steps Through Time, $M^{\textbf{BTT}}$

**Q: How does increasing training-time compute affect performance?**

*SORL generally observes increasing performance with increased training-time compute (i.e increasing $M^{BTT}$), up to a performance saturation point of $M^{BTT} = 4$.*

One of SORL's unique strengths is its scalability at both training-time and inference-time. In this experiment, we consider the question of how much training-time compute is necessary. Based on the results in Figure 4, we see that, in general, performance improves with increased training-time compute (i.e. with an increasing number of backpropagation steps through time, $M^{\text{BTT}}$). However, performance seems to saturate around $M^{\text{BTT}} = 4$, suggesting that greater training-time compute may not be necessary for the environments considered.

The results suggest two key takeaways. First, backpropagation through time over more than one step *is* generally necessary in SORL to maximize performance. Insufficient training-time compute may lead to sub-optimal results, as evidenced in Figure 4 and Tables 1and2, where SORL with $M^{\text{BTT}} \geq 4$ outperforms both the baselines and the versions of SORL with $M^{\text{BTT}} < 4$. Second, the results indicate that SORL does *not* require backpropagation through tens or hundreds of steps—as is common in large diffusion models [Ho et al., 2020, Song et al., 2020]—a procedure that could be computationally prohibitive.

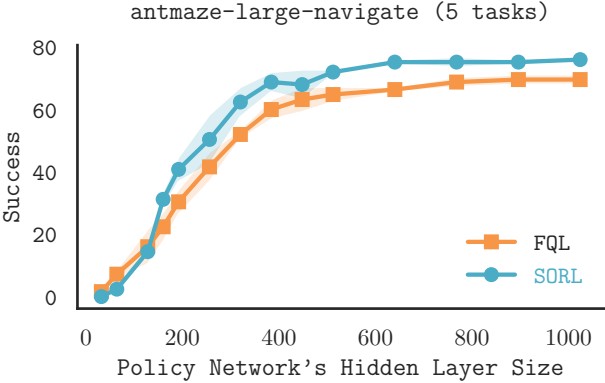

Figure 5: **Ablation Over Policy Network Size.** The performance is averaged over 5 seeds for each task, with 5 tasks per environment, and standard deviations reported. We use the same training-time and inference-time compute budgets for SORL as we use for Tables 1 and 2 (i.e. $M^{\text{BTT}} = 8$ and $M^{\text{inf}} = 4$). We train and evaluate FQL with the parameters used in the official implementation [Park et al., 2025]. The only change to SORL and FQL is varying the sizes of the policy network's hidden layers.

## E.2 Policy Network Size

**Q: How does the performance of SORL vary with changing size of the policy network?**

*Increasing the capacity of the policy network increases the performance of SORL and FQL similarly.*

We use the same training-time and inference-time compute budgets for SORL as we use for Tables 1 and 2 (i.e. $M^{\text{BTT}} = 8$ and $M^{\text{inf}} = 4$). We train and evaluate FQL with the parameters used in the official implementation [Park et al., 2025]. The only change to SORL and FQL is varying the sizes of the policy network's hidden layers. We use an MLP with four hidden layers of the same size. Since FQL uses two actor networks of the same size in the official implementation [Park et al., 2025]—a flow-matching network for offline data regularization and a one-step distillation network for the actor's policy—we scale both networks, keeping them the same size for this ablation study.

As may be expected, increasing the capacity of the policy network increases the performance of both SORL and FQL, at similar rates, before plateauing at larger sizes. Notably, SORL achieves slightly better gains across network sizes, which may stem from the shortcut model class being more expressive than FQL's one-step distillation policy.

# F  Experimental and Implementation Details

In this section, we describe the setup, implementation details, and baselines used in the paper.

## F.1  Experimental Setup

The experimental setup for our paper's main results (Table 1 and 2) follows OGBench's official evaluation scheme [Park et al., 2024a, 2025].

**Environments.**   We now describe the environments, tasks, and offline data used in our experiments. Our setup follows OGBench's official evaluation scheme [Park et al., 2024a] and Park et al. [2025], and we restate the description of environments and tasks below.

We evaluate `SORL` on 8 robotics locomotion and manipulation robotics environments in the OGBench task suite (version `1.1.0`) [Park et al., 2024a], a benchmark suite designed for offline RL. Specifically, we use the following environments:

1. `antmaze-large-navigate-singletask-v0`
2. `antmaze-giant-navigate-singletask-v0`
3. `humanoidmaze-large-navigate-singletask-v0`
4. `humanoidmaze-giant-navigate-singletask-v0`
5. `antsoccer-arena-navigate-singletask-v0`
6. `cube-single-play-singletask-v0`
7. `cube-double-play-singletask-v0`
8. `scene-play-singletask-v0`

The selected environments span a range of challenging control problems, covering both locomotion and manipulation. The `antmaze` and `humanoidmaze` tasks involve navigating quadrupedal (8 degrees of freedom (DOF)) and humanoid (21 DOF) agents through complex mazes. `antsoccer` focuses on goal-directed ball manipulation using a quadrupedal agent. The `cube` and `scene` environments center on object manipulation with a robot arm. Among these, `scene` tasks require sequencing multiple subtasks (up to 8 per episode), while `puzzle` emphasizes generalization to combinatorial object configurations. All environments are state-based. We follow the standard dataset protocols (`navigate` for locomotion, `play` for manipulation), which are built from suboptimal, goal-agnostic trajectories, and therefore pose a challenge for goal-directed policy learning. We evaluate agents using binary task success rates (i.e., goal completion percentage), which is consistent with OGBenchs evaluation setup [Park et al., 2024a].

**Tasks.**   We use OGBench's reward-based `singletask` variants for all experiments [Park et al., 2024a], which are best suited for reward-maximizing RL. As described in the official implementation [Park et al., 2024a], each OGBench environment offers five unique tasks, each associated with a specific evaluation goal, denoted by suffixes `singletask-task1` through `-task5`. These represent unique, fixed goals for the agent to accomplish. We utilize all five tasks for each environment. The datasets are annotated with a semi-sparse reward, where the reward is determined by the number of remaining subtasks at a given step for manipulation tasks, or whether the agent reaches a terminal goal state for locomotion tasks [Park et al., 2024a, 2025].

**Evaluation.**   We follow OGBench's official evaluation scheme [Park et al., 2024a]. We train algorithms for 1,000,000 gradient steps and evaluate 50 episodes every 100,000 gradient steps. We report the average success rates of the final three evaluations (i.e. the evaluation results at 800,000, 900,000, and 1,000,000 gradient steps). In tables, we average over 8 seeds per task and report standard deviations, bolding values within 95% of the best performance.

### F.2 `SORL` Implementation Details

One of the strengths of `SORL` is its implementation simplicity. In order to implement `SORL`, we adapt Park et al. [2025]'s open-source implementations of various offline RL algorithms (FQL, IFQL, IQL, ReBRAC), which are adapted from Park et al. [2024a]'s open-source dataset and codebase. The major changes are implementing and training the shortcut model, and removing additional training complexity from FQL (e.g. the teacher/student networks). To ensure a fair comparison, we keep all major shared hyperparameters the same between the baselines and `SORL` (e.g. network size, number of gradient steps, and discount factor), unless otherwise noted.

**Value Network.** We train two $Q$ functions and use the mean of the two in the actor update (Equation 9). We use the mean of the two $Q$ functions for the critic update (Equation 14), except for `antmaze-large` and `antmaze-giant` tasks, where we follow Park et al. [2025] in using the minimum of the two values. The only change to the FQL baseline's value network is adding an input that encodes the number of inference steps used to generate the actions.

**Network Architecture and Optimizer.** We use a multi-layer perceptron with 4 hidden layers of size 512 for both the value and policy networks. We apply layer normalization [Ba et al., 2016] to value networks. We use the Adam optimizer [Kingma, 2014], which we add gradient clipping to.

**Discretization Steps.** For `SORL`, we use $M^{\text{disc}} = 8$ discretization steps during training (2 fewer than FQL), since Frans et al. [2024] suggests that discretization steps be powers of 2. By default for the results in Tables 1 and 2, we use $M^{\text{BTT}} = 8$ steps of backpropagation through time and $M^{\text{inf}} = 4$ inference steps, except for `humanoidmaze-medium` where we use $M^{\text{inf}} = 2$. In other experiments, if the values of $M^{\text{BTT}}$ and $M^{\text{inf}}$ are changed, they are explicitly noted.

**Shortcut Model.** Following the official implementation of shortcut models [Frans et al., 2024], for the self-consistency loss we sample $d \sim \{2^k\}_{k=0}^{\log_2 M - 1}$ and then sample $t$ uniformly on multiples of $d$ between 0 and 1 (i.e. points where the model may be queried). The model makes a prediction for step size $2d$, and its target is the concatenation of the two sequential steps of size $d$. For greater stability, we construct the targets via a target network. Unlike Frans et al. [2024], we do not require special processing of the training batch into empirical and self-consistency targets, nor do we require special weight decay. We simply use the entire batch for the critic and actor updates, including the $Q$ loss, self-consistency loss, and flow-matching loss.

**Hyperparameters.** We largely use the same hyperparameters for `SORL` as those used in the FQL baseline (Table 3), except for parameters that are specific to or vary with `SORL` (Table 4). We hyperparameter tune `SORL` with a similar training budget to the baselines: we only tune one of `SORL`'s training parameters—Q-loss (QL) coefficient—on the *default* task in each environment (Table 5), over the values {10, 50, 100, 500}. We then use the same QL coefficient on all the tasks in an environment. Varying the strength of regularization to the offline data is common in offline RL [Tarasov et al., 2023b, Park et al., 2024b, 2025]. Following Park et al. [2025]'s recommendation, we normalize the Q loss.

Table 3: Shared Hyperparameters Between FQL Baseline and SORL.

| PARAMETER | VALUE |
|---|---|
| OPTIMIZER | ADAM [KINGMA, 2014] |
| GRADIENT STEPS | 1,000,000 |
| MINIBATCH SIZE | 256 |
| MLP DIMENSIONS | [512, 512, 512, 512] |
| NONLINEARITY | GELU [HENDRYCKS AND GIMPEL, 2016] |
| TARGET NETWORK SMOOTHING COEFFICIENT | 0.005 |
| DISCOUNT FACTOR $\gamma$ | 0.99 (DEFAULT), 0.995 (ANTMAZE-GIANT, HUMANOIDMAZE, ANTSOCCER) |
| DISCRETIZATION STEPS | 8 |
| TIME SAMPLING DISTRIBUTION | UNIF([0,1]) |
| CLIPPED DOUBLE Q-LEARNING | FALSE (DEFAULT), TRUE (ADROIT, ANTMAZE-{LARGE, GIANT}-NAVIGATE) |

Table 4: Hyperparameters for SORL.

| HYPERPARAMETER | VALUE |
|---|---|
| LEARNING RATE | 1E-4 |
| GRADIENT CLIPPING NORM | 1 |
| DISCRETIZATION STEPS | 8 |
| BC COEFFICIENT | 10 |
| SELF-CONSISTENCY COEFFICIENT | 10 |
| Q-LOSS COEFFICIENT | TABLE 5 |

Table 5: Q-Loss Coefficient for SORL.

| ENVIRONMENT | Q LOSS COEFFICIENT |
|---|---|
| antmaze-large-navigate-v0 (5 tasks) | 500 |
| antmaze-giant-navigate-v0 (5 tasks) | 500 |
| humanoidmaze-medium-navigate-v0 (5 tasks) | 100 |
| humanoidmaze-large-navigate-v0 (5 tasks) | 500 |
| antsoccer-arena-navigate-v0 (5 tasks) | 500 |
| cube-single-play-v0 (5 tasks) | 10 |
| cube-double-play-v0 (5 tasks) | 50 |
| scene-play-v0 (5 tasks) | 100 |

### F.3 Baselines

We evaluate against three Gaussian-based offline RL algorithms (`BC` [Pomerleau, 1988], `IQL` [Kostrikov et al., 2021], `ReBRAC` [Tarasov et al., 2023a]), three diffusion-based algorithms (`IDQL` [Hansen-Estruch et al., 2023], `SRPO` [Chen et al., 2023], `CAC` [Ding and Jin, 2023]), and four flow-based algorithms (`FAWAC` [Nair et al., 2020], `FBRAC` [Zhang et al., 2025], `IFQL` [Wang et al., 2022], `FQL` [Park et al., 2025]). For the baselines we compare against in this paper, we report results from Park et al. [2025], who performed an extensive tuning and evaluation of the aforementioned baselines on OGBench tasks. We restate descriptions of the baselines here.

**Gaussian-Based Policies.** To evaluate standard offline reinforcement learning (RL) methods that employ Gaussian policies, we consider three representative baselines: Behavior Cloning (`BC`)[Pomerleau, 1988], Implicit Q-Learning (`IQL`) [Kostrikov et al., 2021], and `ReBRAC` [Tarasov et al., 2023a]. `BC` [Pomerleau, 1988] serves as a simple imitation learning baseline that directly mimics the demonstration data without any explicit value optimization, while `IQL` [Kostrikov et al., 2021] represents a popular value-based approach that addresses the overestimation problem in offline RL through implicit learning of the maximum value function. `ReBRAC` [Tarasov et al., 2023a], a more recent method that is known to perform well on many D4RL tasks [Tarasov et al., 2023b], extends the `BRAC` framework with regularization techniques specifically designed to constrain the learned policy close to the behavior policy, thereby mitigating the distributional shift problem common in offline RL settings.

**Diffusion-Based Policies.** To evaluate diffusion policy-based offline RL methods, we compare to `IDQL` [Hansen-Estruch et al., 2023], `SRPO` [Chen et al., 2023] and Consistency-AC (`CAC`) [Ding and Jin, 2023]. `IDQL` [Hansen-Estruch et al., 2023] builds on `IQL` by using a generalized critic and rejection sampling from a diffusion-based behavior policy. `SRPO` [Chen et al., 2023] replace the diffusion sampling with a deterministic policy trained via score-regularized policy gradient to speed up sampling. `CAC` [Ding and Jin, 2023] introduces a consistency-based actor-critic framework to backpropagation through time with fewer steps.

**Flow-Based Policies.** To evaluate flow policy-based offline RL methods, we compare flow-based variants of prior methods, including new variants introduced by Park et al. [2025]: `FAWAC` [Nair et al., 2020], `FBRAC` [Wang et al., 2022], `IFQL` [Hansen-Estruch et al., 2023], and `FQL` [Park et al., 2025], as only a few previous works explicitly employ flow-based policies. `FAWAC` extends `AWAC` by adopting a flow-based policy trained with the AWR objective and estimates $Q^\pi$ for the current policy using fully off-policy bootstrapping. `FBRAC` is the flow counterpart of Diffusion-QL, based on the naïve Q-loss with backpropagation through time. `IFQL` is a flow-based variant of `IDQL` that relies on rejection sampling. `FQL` distills a one-step policy from an expressive flow-matching policy to avoid costly iterative sampling. Unlike `FQL`, `SORL` uses a single-stage training procedure and does not require distilling a model into a one-step policy.

