# OpenReview forum: "Scaling Offline RL via Efficient and Expressive Shortcut Models"
_NeurIPS.cc/2025/Conference — NeurIPS 2025 poster_

### Official Review · Reviewer_Ewda · 2025-07-03

**Clarity:** 3
**Significance:** 2
**Originality:** 2
**Rating:** 4
**Confidence:** 4

**Summary:**

This paper studies the problem of applying diffusion and flow models to the offline RL problem. The authors introduce a method called Scalable Offline Reinforcement Learning (SORL) which leverages *shortcut models* (a novel class of generative models) for scalability in training time and inference time. The authors establish theoretical results that the shortcut models output a distribution that is close (in Wasserstein) to the data distribution (and thus, that the offline RL policy is regularized towards the behavior policy). Experiments are conducted.

**Questions:**

1. What is the point of the $Q$ loss $L_{QL}(\theta)$ in Eq. 9?
2. Can you provide more analysis of why certain environments didn't benefit from this approach?
3. What are the actual benefits of this method in terms of total wall-clock time?

**Ethical Concerns:**

["NO or VERY MINOR ethics concerns only"]

**Limitations:**

See above

**Quality:**

3

**Strengths And Weaknesses:**

At a high level, this paper amounts to "importing" shortcut models into RL by using them as a policy class in an otherwise standard RL pipeline. Despite the lack of algorithmic novelty, the approach is well-motivated and the authors are convincing about the benefits of this approach, namely that it can simultaneously give expressive multi-modal policies, provide efficient training, and achieve inference-time training. The experimental results are mixed but mostly strong and the theoretical analysis of shortcut models is novel (from what I can tell) and may be of independent interest. The shortcut model analysis itself is not a complete analysis of the SORL method (it only analyses the regularization term by itself). For the experimental results, it would have been more convincing if total wall-clock time was measured, since the authors claim that faster training and inference is possible with this method.

Typos:
1. Algorithm 2: $n$ should be $t$
2. Critic loss is missing an expectation over data
3. Line 192: "we prove that the is *yes*."

---

> ### Author Rebuttal · Authors · 2025-07-30
>
> # 1. Typos
> > Algorithm 2: n should be t
>
> > Critic loss is missing an expectation over data
>
> > Line 192: "we prove that the is yes."
>
> - Thank you for the comments. We will update the paper accordingly.
>
> # 2. Q Update in Eq. 9
> > What is the point of the loss in Eq. 9?
> - The **purpose of the Q loss is to update the policy in a way that selects actions with a higher Q value** (that is, a higher expected return). This is the “RL optimization” component of our loss. The flow matching loss serves as our “regularizer” to the offline data.
>
> # 3. Analysis of Performance
> > Can you provide more analysis of why certain environments didn't benefit from this approach?
> - **The key advantage of SORL over Gaussian-based methods (e.g. ReBRAC) is its more expressive policy class.** This policy class is better able to handle diverse, multi-modal data than Gaussian-based models, which cannot handle multi-modality. The key advantage of SORL over distillation-based generative model approaches (e.g. SRPO, CAC, FQL) is SORL’s one-stage policy training procedure, as compared to the two-stage distillation-based approaches over which error compounds. **SORL’s flexibility in selecting the number of discretization and inference steps allows for less discretization error than a one-step approach**. We believe this contributes to more precise actions at test-time.
> - The environments where SORL struggled (antmaze-giant and humanoidmaze-large) are very challenging tasks in offline RL, and it should be noted that none of the 10 baselines we considered performed well on these environments. SORL always achieved at least the second best performance among the 10 baselines considered. On the two environments where SORL does not achieve the best performance among the baselines, there is only one algorithm that outperforms it (ReBRAC on antmaze-giant and IFQL on humanoidmaze-large). Both ReBRAC and IFQL fail to achieve the best performance on any other environment, highlighting the difficulty of having reaching optimal performance on every environment.
> - At a higher-level, the antmaze-giant and humanoidmaze-large tasks are known to be challenging, long-horizon environments, and **it is possible that these environments would benefit significantly from online exploration**. As noted in the limitations discussed in Appendix B, SORL does not have a principled exploration strategy, which is standard for _offline_ algorithms. Depending on the coverage and quality of the offline dataset, purely offline learning may get stuck in local optima.
> - Relatedly, **it is possible that policy extraction is not the primary bottleneck in these two environments** under the given dataset size (1 million transitions). It may be the case that, for these two environments, value learning is challenging. If the learned value function (e.g. Q function) is poor, then it will be difficult to optimize a policy under that value function, irrespective of the policy extraction method, expressivity of the policy class, or discretization/integration error.
>
> # 4. Runtime
> > What are the actual benefits of this method in terms of total wall-clock time?
> - In Appendix C.2, **we provided a runtime comparison against the fastest baseline** FQL [Park et al., 2025], which **we decompose into training and inference**. We show that SORL offers flexibility under varying compute budgets: SORL can take advantage of greater computation during both training and inference, or opt for a faster training/inference procedure.

---

> > ### Author Response · Authors · 2025-08-05
> >
> > Hi Reviewer Ewda, thank you again for your feedback. We’ve done our best to respond to the points you raised in the review. If there’s anything we haven't addressed to your satisfaction, we’d be happy to clarify further. If you feel that our response resolves your concerns, we'd kindly ask you to consider updating your score. Thank you again for your time and feedback.

---

### Official Review · Reviewer_3Dwq · 2025-07-03

**Clarity:** 3
**Significance:** 4
**Originality:** 3
**Rating:** 5
**Confidence:** 3

**Summary:**

This paper proposes SORL, a new offline RL algorithm that leverages expressive generative models as a policy class and allows for direct optimisation of the policy with the gradients of the critic using shortcut models. This avoids the instability of back propagation through time (BPTT) that exists when optimizing vanilla flow/diffusion models with RL due to the iterative denoising process. Using shortcut models allows for using different denoising steps for the policy optimisation and the behaviour cloning terms, achieving the best of both paradigms. It also means that practitioners can scale inference time compute when available, using more steps when precision is important, or fewer steps in compute constrained or latency sensitive environments (like edge deployments). They conduct extensive evaluations to show how SORL achieves better performance compared to existing baselines and also validate their inference time scaling hypothesis. That is, allowing for more denoising steps at inference generally leads to better performance. This is in line with the recent literature studying test time scaling in LLMs and is very useful for practical deployments of offline RL methods.

**Questions:**

- I am unsure of the range of $h$ that is used to define the self consistency loss in Eq 5. It seems to be used to both define the number of steps ($\in [1, M]$)  as well as the dt time interval ($\in [0, 1]$) that the shortcut model is conditioned on. The text mentions (in L110) that $h \in [1, M]$ and in L111 $h \in ( 2^k )_{k=0}^{log_2M}$, where $M$ is the number of discretisation steps. But then shouldn't the $h$ in the flow matching part of $\mathcal{L}^S$ be 1 and not $1/M$? Since the flow matching loss is for one step. But L112 defined the flow loss as when $h=1/M$ meaning h refers to the dt?
- In Algorithm 1 is there a reason that you compute the component losses of the actor on separate sections of the batch? That is the Q maximisation loss is only on the first third, the self consistency only on the second third and so on?
- Also in Algorithm 1, isn't the sign of the grad of the Q term inverted, and it is missing the flow matching term too I think?
- It might be easier to understand Fig 2 if the x axis was grouped by $M^{BTT}$. That way the gains from increasing $M^{inf}}$ for a fixed training budget (fixed $M^{BTT}$ would be easier to see. Maybe the figure could be split into two, one which tests for scaling inference steps for lower training $M^{BTT}$ and another which tests generalization to inference steps beyond what was used in training.

**Ethical Concerns:**

["NO or VERY MINOR ethics concerns only"]

**Final Justification:**

The paper is well motivated and presents strong results. Integrating RL with expressive generative models is of relevance to the RL and robotics community. For these reasons, I believe the paper should be accepted.

**Limitations:**

Yes

**Quality:**

4

**Strengths And Weaknesses:**

Strengths
- Comprehensive evaluation against relevant baselines. The paper presents results on a variety of control tasks and show that SORL beats strong baselines in challenging tasks.
- The paper validates the claim of inference time scaling, with experiments showing better performance as $M^{inf}$ is increased.
- Shortcut models preserve the expressivity of the network while also being suitable for RL optimisation without needing a separate distillation step as in FQL.
- The method is well motivated and easy to implement.

Weaknesses
- There are some notations and figures which are unclear. See questions below.

Overall I have no major concerns with the paper.

---

> ### Author Rebuttal · Authors · 2025-07-30
>
> We thank the reviewer for their comments and respond to their questions below.
>
> # 1. Questions
> > I am unsure of the range of $h$ that is used to define the self consistency loss in Eq 5. It seems to be used to both define the number of steps ($\in [0, 1]$) as well as the dt time interval ($\in [0, 1]$) that the shortcut model is conditioned on. The text mentions (in L110) that $h \in [1, M]$ and in L111 $h \in \left ( 2^k \right )^{\log_2 M}_{k=0}$, where $M$ is the number of discretisation steps. But then shouldn't the $h$ in the flow matching part of $\mathcal{L}^S$ be 1 and not $1/M$? Since the flow matching loss is for one step. But L112 defined the flow loss as when $h=1/M$ meaning h refers to the dt?
> - Thank you for the comment. We will correct the definition of shortcut models to avoid overloading notation and avoid the confusion. The last parameter passed into the shortcut model s_theta should be the _step size_.
>
> > In Algorithm 1 is there a reason that you compute the component losses of the actor on separate sections of the batch? That is the Q maximisation loss is only on the first third, the self consistency only on the second third and so on?
> - In practice, all components in our implementation use the same batch. The separate data samples in Algorithm 1 was a mistake in our pseudocode. We will update the algorithm’s pseudocode to reflect the implementation.
>
> > Also in Algorithm 1, isn't the sign of the grad of the Q term inverted, and it is missing the flow matching term too I think?
> - Yes, we will update the expression accordingly.
>
> > It might be easier to understand Fig 2 if the x axis was grouped by $M^{\text{BTT}}$. That way the gains from increasing $M^{\text{inf}}$ for a fixed training budget (fixed $M^{\text{BTT}}$ would be easier to see. Maybe the figure could be split into two, one which tests for scaling inference steps for lower training and another which tests generalization to inference steps beyond what was used in training.
> - Thank you for the suggestion. We will update the plots accordingly.

---

### Official Review · Reviewer_rsZE · 2025-07-03

**Clarity:** 3
**Significance:** 2
**Originality:** 2
**Rating:** 4
**Confidence:** 3

**Summary:**

The authors introduce Scalable Offline Reinforcement Learning (SORL), a new offline RL algorithm that leverages shortcut models to scale both training and inference. SORL's policy can be trained efficiently in a one-stage training procedure. At test time, SORL supports both sequential and parallel inference scaling by using the learned Q-function as a verifier. Experiments are conducted to validate the effectiveness of the proposed method.

**Questions:**

Apart from other weaknesses, I list my confusion about the equations in weakness 3 here.

1. In Eq. 7, the minus sign should be plus sign?

2. In Eq. 2, is the trivial solution $f=z_1-z_0$? This makes the objective 0, achieving the smallest value (since $||\cdot||_2>0$).

**Ethical Concerns:**

["NO or VERY MINOR ethics concerns only"]

**Final Justification:**

I have read the author's response. For the novelty, inference-time scaling is also an existing method. However, the combination of the short-cut model and the inference-time scaling seems ok. I will increase my score to 4.

**Limitations:**

yes

**Quality:**

3

**Strengths And Weaknesses:**

**Strengths:**
1. The scaling of the (offline) RL method is an important and interesting problem.
2. The paper is well written.

**Weaknesses:**
1. Straightforward Theory. It seems like the conclusion that Theorem 2 tries to prove is trivial: due to the flow matching loss in Eq.2, the policy is learned with regularization, and hence, SORL can learn a policy that is regularized to the behavior of the offline data. It seems like a standard offline RL approach.

2. Lacking Training Scalibility Experiments. The authors claim the proposed method scales both training and inference in the paper. Yet, the training scalability experiments are missing.

3. Some Equations Flaws. See the question part.

4. Novelty concern. The paper employs an existing type of generative model, the shortcut model (proposed around 2023), to implement the policy in offline RL. Is this enough to construct a novel idea or just a model implementation trick?

---

> ### Author Rebuttal · Authors · 2025-07-30
>
> # 1. Theory
> > Straightforward Theory. It seems like the conclusion that Theorem 2 tries to prove is trivial: due to the flow matching loss in Eq.2, the policy is learned with regularization, and hence, SORL can learn a policy that is regularized to the behavior of the offline data. It seems like a standard offline RL approach.
> - We agree with the reviewer that the ultimate aim of Theorem 2 involves a standard one for offline RL and generative model theory: proving that minimizing the training objective keeps the learned policy “close” to the behavior policy, where closeness is measured by Wasserstein distance.
> - However, we emphasize two points: **1) our analysis is an important theoretical grounding for our algorithm**, and **2) our analysis of shortcut models is novel**. While the overall aim of our theoretical results may be standard, we are using a novel class of generative model that has not been used before in offline RL. Moreover, the paper that introduced shortcut models [Frans et al., 2024] lacked any theoretical results, **making our theoretical analysis of shortcut models the first to our knowledge**.
> - More specifically, Theorems 2 and 3 involve the convergence of the shortcut model. The proof of the shortcut model’s convergence is non-trivial because of distributional shift: the flow-matching loss and the self-consistency loss are trained on the _ground-truth_ marginal distribution, not the marginal distribution _during training_. This distributional shift happens at the level of the shortcut model’s inference step, which is separate from the standard offline RL-notion of distributional shift, which occurs at the the MDP-level. The distributional shift we analyze is further complicated because of the self-consistency step in shortcut models. In summary, we argue that the our theoretical analysis of shortcut models is a novel and important contribution to the theoretical grounding of our algorithm.
>
> # 2. Training Scalability
> > Lacking Training Scalability Experiments. The authors claim the proposed method scales both training and inference in the paper. Yet, the training scalability experiments are missing.
> - As discussed in the paper, there are two principal ways of scaling training. First, in order to scale training, offline RL algorithms must handle large, diverse, multi-modal datasets. **To demonstrate SORL’s training scalability, we test it on a variety of sub-optimal datasets and offline RL tasks**. We show that, for a fixed dataset, SORL can generally use the data to achieve a higher performance than the other 10 baseline methods considered. The second avenue to scale training is focused on computation (i.e. how much computation is required to train an algorithm.) We follow prior work in training SORL (and all baselines) for 1M gradient steps. **We then provided a runtime comparison to the fastest baseline in Appendix C.2**.  We show that SORL is flexible: it can be optimized for fast training, as measured by training runtime, by decreasing the number of backpropagation steps through time, $M^{\text{BTT}}$. And, SORL can be optimized to leverage greater computation during training to further increase performance, by increasing the number of backpropagation steps through time, $M^{\text{BTT}}$.
>
> # 3. Novelty
> > Novelty concern. The paper employs an existing type of generative model, the shortcut model (proposed around 2023), to implement the policy in offline RL. Is this enough to construct a novel idea or just a model implementation trick?
> - Prior work [Hansen-Estruch et al., 2023, Chen et al., 2023, Ding and Jin, 2023, Wang et al., 2022, Park et al., 2025] focus on scaling training in offline RL by leveraging expressive policy classes that can model complex data distributions. As the reviewer points out, part of our paper follows a similar approach, leveraging shortcut models as the new type of generative model for offline RL training. However, **our paper extends beyond a simple model implementation trick by tackling the question of inference-time scaling in offline RL with generative models**. Motivated by the recent work in language model test-time scaling, SORL introduces methods for sequential and parallel scaling which enable flexible inference: SORL can perform both fast inference or high precision inference, depending on the test-time compute budget, without having to be retrained. We argue that introducing avenues for test-time scaling in offline RL presents a novel contribution unaddressed by prior work.
>
> # 4. Equations
> > In Eq. 7, the minus sign should be plus sign?
> - Yes, the minus sign should be plus. We will fix this mistake.
>
> > In Eq. 2, is the trivial solution $f=z_1 - z_0$? This makes the objective 0, achieving the smallest value (since $\|\cdot \|_2 > 0$).
> - To clarify, the function $f$ in Eq. 2 takes $(x_t, t)$ as inputs, not $(x_0, x_1)$. The setting we consider here is standard see Liu et al. [2022].

---

> > ### Comment · Reviewer_rsZE · 2025-08-04
> >
> > Thank you for the response. For the novelty, inference-time scaling is also an existing method. However, the combination of the short-cut model and the inference-time scaling seems ok. I will increase my score to 4.

---

### Official Review · Reviewer_VRv3 · 2025-07-06

**Clarity:** 3
**Significance:** 2
**Originality:** 2
**Rating:** 4
**Confidence:** 5

**Summary:**

The paper proposes a method to train a shortcut model for offline RL tasks. The training loss consists of three components: flow-matching loss, consistency loss, and Q loss. Unlike two-stage distillation frameworks such as FQL, the proposed method (SORL) is a single-stage approach that offers good scalability at both training and test time.

**Questions:**

- **Loss components and data usage (Alg. 1)**: Why does the algorithm use different data samples for computing different loss components? Is there any benefit to this? Would it cause issues if all components used the same batch?

- **Critic update with sampled inference steps**: During critic updates, the algorithm samples $m$ inference steps. When $m$ is small, the resulting actions might be noisy. Could this hurt Q-estimation accuracy and affect Bellman equation convergence—especially if the shortcut model is not yet well trained?

- **Equation 14 and min-Q trick**: I noticed from the appendix that the method uses the min-Q trick. Please consider integrating this directly into Equation 14 for better clarity.

- **Figure 1 (AntMaze-large)**: Why does increasing inference steps lead to worse performance on AntMaze-large tasks?

- **Table 2 (AntMaze-giant)**: SORL performs poorly on AntMaze-giant. It seems that generative RL methods struggle on this task. Any insights into why this might be?

- **Figure 3 (Scalability benefits)**: Does Figure 3 suggest that training-time scalability is more beneficial than test-time scalability? What happens if you increase $M^{\text{BTT}}$ to 16? Does performance continue to improve?

**Ethical Concerns:**

["NO or VERY MINOR ethics concerns only"]

**Final Justification:**

I have no concerns remained and decide to keep my score as 4.

**Limitations:**

It would be promising to see the proposed method tested on more high-dimensional offline RL tasks to further validate its scalability.

**Quality:**

3

**Strengths And Weaknesses:**

**Strengths:**

- The question of training one-step generative models directly for rl setting (rather than relying on two-stage frameworks) is timely and important.
- The method is well designed. By incorporating Q-loss into shortcut model training, it offers scalability in both training and inference.
- The theoretical analysis appears sound with no obvious flaws. The experiments are standard and show strong performance compared to recent baselines like FQL.

**Weaknesses:**

- Given that scalability is central to the paper, it's worth asking whether the method has been tested on higher-dimensional inputs (e.g., image-based observations). That would further support the scalability claims.

---

> ### Author Rebuttal · Authors · 2025-07-30
>
> We thank the reviewer for their comment. We respond to their questions and concerns below.
>
> # 1. Higher-Dimensional Inputs
> > Given that scalability is central to the paper, it's worth asking whether the method has been tested on higher-dimensional inputs (e.g., image-based observations). That would further support the scalability claims.
> - Image-based observations are out of the scope of our paper. The **focus of our paper is on policy extraction and inference-time scaling**. Using image-based observations would require an encoder, the choice of which would significantly impact performance.
>
> # 2. Loss Components and Data Usage
> > Loss components and data usage (Alg. 1): Why does the algorithm use different data samples for computing different loss components? Is there any benefit to this? Would it cause issues if all components used the same batch?
> - In practice, **all components in our implementation use the same batch**. The separate data samples in Algorithm 1 was a mistake in our pseudocode. We will update the algorithm’s pseudocode to reflect the implementation.
>
> # 3. Critic Update
> > Critic update with sampled inference steps: During critic updates, the algorithm samples $m$ inference steps. When $m$ is small, the resulting actions might be noisy. Could this hurt Q-estimation accuracy and affect Bellman equation convergence—especially if the shortcut model is not yet well trained?
> - **By enforcing self-consistency during training, we ensure that sampling with smaller $m$, including the 1-step policy, achieves a discretization error comparable to the Euler method with larger $m$**. The reviewer’s question points to a common issue faced by actor-critic style methods: both the actor and critic updates depend on each other’s outputs, and they will have to handle "noisy" outputs at the start of training. We mitigate this issue by using target networks for both the actor and critic, which ensures a “slower update” for both the actor and critic. As discussed in Appendix E.2, SORL does not require significant hyperparameter tuning beyond the standard regularization parameter in offline RL.
>
> # 3. Min-Q Trick
> > Equation 14 and min-Q trick: I noticed from the appendix that the method uses the min-Q trick. Please consider integrating this directly into Equation 14 for better clarity.
> - Thank you for the comment, and we will update Equation 14 accordingly.
>
> # 4. Figure 1 (AntMaze-Large)
> > Figure 1 (AntMaze-large): Why does increasing inference steps lead to worse performance on AntMaze-large tasks?
> - **The reason for the sequential scaling trend in antmaze-large remains unclear**, but notably, **SORL achieves the best performance of any method under every number of inference steps**, and they are all a substantial improvements over the 10 baselines. The difference in SORL’s performance under the varying inference steps is smaller compared to other environments.
>
> # 5. Table 2 (AntMaze-Giant)
> > Table 2 (AntMaze-giant): SORL performs poorly on AntMaze-giant. It seems that generative RL methods struggle on this task. Any insights into why this might be?
> - Antmaze-Giant is a very challenging task in offline RL, and it should be noted that **SORL achieves the second-best performance on this environment**. Additionally ReBRAC fails to achieve the best performance on any other environment, highlighting the difficulty of having reaching optimal performance on every environment.
> - More generally, the key advantage of SORL and other generative model-based approaches over Gaussian-based methods (e.g. ReBRAC) is its more expressive policy class. This policy class is better able to handle diverse, multi-modal data than Gaussian-based models, which cannot handle multi-modality. The key advantage of SORL over distillation-based generative model approaches (e.g. SRPO, CAC, FQL) is SORL’s one-stage policy training procedure, as compared to the two-stage distillation-based approaches over which error compounds. SORL’s flexibility in selecting the number of discretization and inference steps allows for less discretization error than a one-step approach. We believe this contributes to more precise actions at test-time.
> - **It is possible that policy extraction is not the primary bottleneck in these two environments** under the given dataset size (1 million transitions). It may be the case that, for these two environments, value learning is challenging. If the learned value function (e.g. Q function) is poor, then it will be difficult to optimize a policy under that value function, irrespective of the policy extraction method, expressivity of the policy class, or discretization/integration error.
>
> # 6. Figure 3 (Scalability Benefits)
> > Figure 3 (Scalability benefits): Does Figure 3 suggest that training-time scalability is more beneficial than test-time scalability? What happens if you increase
> - **The question of whether training-time or test-time scalability is more beneficial is difficult to answer because it relies on a tradeoff between compute and performance**. For some settings, test-time compute may be low because the robot must perform inference quickly (e.g. autonomous vehicles), while in other tasks, precision may be valued over speed (e.g. surgical robotics).

---

> > ### Author Response · Authors · 2025-08-05
> >
> > Hi Reviewer VRv3, thank you again for your thoughtful feedback. We’ve tried to thoroughly respond to your review's comments. If there’s anything we haven't sufficiently addressed, we’d be grateful for the chance to clarify further. If you feel that our response resolves your concerns, we'd kindly ask you to consider updating your score. Thank you again for your time and feedback.

---

> > ### Comment · Reviewer_VRv3 · 2025-08-05
> >
> > Thank you to the authors for the detailed response. My concerns have been addressed, and I have decided to keep my current rating.

---

### Official Review · Reviewer_GJHL · 2025-07-06

**Clarity:** 2
**Significance:** 3
**Originality:** 3
**Rating:** 4
**Confidence:** 3

**Summary:**

This paper presents Scalable Offline Reinforcement Learning (SORL), an efficient offline RL algorithm that leverages shortcut models to scale training and inference. SORL incorporates self-consistency into training, enabling it to optimize policies with fewer backpropagation steps while maintaining high expressivity. It achieves strong performance across diverse tasks and can dynamically adjust inference steps based on compute budget, demonstrating improved performance with increased test-time compute.

**Questions:**

1.	How does SORL perform on more complex or diverse tasks beyond the current environments tested? The current results might not reflect its performance on a broader range of real-world scenarios.
2.	What are the specific factors that contribute to SORL’s superior performance in some environments but not in others? Understanding these factors could provide deeper insights into the algorithm’s strengths and weaknesses.
3.	Can the authors provide more detailed analysis or additional experiments to further validate the robustness and generalizability of SORL’s performance? More comprehensive testing might be needed to fully assess its potential.
4.	While SORL aims to improve training efficiency, what is the trade-off in terms of computational resources required during inference? Does the potential for test-time scaling compensate for any increased computational demands?
5.	How does SORL compare to other recent advances in offline RL, particularly those that also focus on scaling or improving inference efficiency? Are there any key works that should be included in the discussion to provide a more comprehensive comparison?
6.	What are the potential limitations of using shortcut models in SORL, especially in terms of capturing long-term dependencies or handling highly complex data distributions?

**Ethical Concerns:**

["NO or VERY MINOR ethics concerns only"]

**Limitations:**

Yes

**Quality:**

3

**Strengths And Weaknesses:**

Strengths：
1.	The paper tackles the significant challenge of scaling offline reinforcement learning, which is crucial for practical applications involving large datasets and complex environments.
2.	The authors present their ideas and methods clearly, making the paper accessible and easy to understand, even for those new to the field.
3.	The paper includes a thorough set of experiments across diverse tasks, demonstrating the robustness and effectiveness of the proposed method.

Weaknesses:
1.	The experiments are conducted on a relatively narrow set of environments (8 environments with 40 tasks in total). This might not fully capture the breadth of challenges in offline RL, limiting the generalizability of the results.
2.	While the paper compares SORL to 10 baselines, the results show that SORL only outperforms them in 5 out of 8 environments. This suggests that its superiority might not be consistent across all types of tasks.
3.	The paper may not provide a comprehensive review of existing methods and their limitations, which is crucial for understanding how SORL fits into the broader context of offline RL research. A more thorough discussion could highlight the unique contributions of SORL more clearly.

---

> ### Author Rebuttal · Authors · 2025-07-30
>
> We thank the reviewers for their comments. We respond to their questions and concerns below.
>
> # 1. Environments and Tasks
> > The experiments are conducted on a relatively narrow set of environments (8 environments with 40 tasks in total). This might not fully capture the breadth of challenges in offline RL, limiting the generalizability of the results.
>
> > How does SORL perform on more complex or diverse tasks beyond the current environments tested? The current results might not reflect its performance on a broader range of real-world scenarios.
> - **We evaluate SORL against 10 baselines on 40 unique tasks, spanning 8 environments, on a recently released offline RL benchmark [Park et al., 2024a]**.
> - The 8 environments span locomotion and manipulation tasks, including long-horizon tasks (e.g. scene), tasks involving both high-level, goal-directed planning and low-level motor control (e.g. antmaze, humanoidmaze, antsoccer), and tasks involving sub-tasks (cube).
> - Our focus on robotics simulations is standard in the RL literature.
>
> > Can the authors provide more detailed analysis or additional experiments to further validate the robustness and generalizability of SORL’s performance? More comprehensive testing might be needed to fully assess its potential.
> - We emphasize that **our evaluation is performed on a diverse set of tasks and environments** that is either similar to [Park et al., 2025] or more diverse than prior work in offline RL with generative models [Hansen-Estruch et al., 2023, Chen et al., 2024, Ding et al., 2024].
>
> # 2. Baselines
> > While the paper compares SORL to 10 baselines, the results show that SORL only outperforms them in 5 out of 8 environments. This suggests that its superiority might not be consistent across all types of tasks.
> - We emphasize that **SORL always achieved the second-best performance in _every_ environment evaluated**, and it achieved **the best performance on 5 environments (25 tasks)**.
>
> # 3. Review of Existing Methods
> > 3. The paper may not provide a comprehensive review of existing methods and their limitations, which is crucial for understanding how SORL fits into the broader context of offline RL research. A more thorough discussion could highlight the unique contributions of SORL more clearly.
> - In Appendix A, **we provided a discussion of related work**, and in Appendix E.3, **we provided a thorough discussion of all the baselines** we compare to in our paper.
>
> # 4. Factors Affecting SORL's Performance
> > What are the specific factors that contribute to SORL’s superior performance in some environments but not in others? Understanding these factors could provide deeper insights into the algorithm’s strengths and weaknesses.
> - **The key advantage of SORL over Gaussian-based methods (e.g. ReBRAC) is its more expressive policy class**. This policy class is better able to handle diverse, multi-modal data than Gaussian-based models, which cannot handle multi-modality. **The key advantage of SORL over distillation-based generative model approaches (e.g. SRPO, CAC, FQL) is SORL’s one-stage policy training procedure**, as compared to the two-stage distillation-based approaches over which error compounds. **SORL’s flexibility in selecting the number of discretization and inference steps allows for less discretization error** than a one-step approach. We believe this contributes to more precise actions at test-time.
> - The environments where SORL struggled (antmaze-giant and humanoidmaze-large) are very challenging tasks in offline RL, and it should be noted that none of the 10 baselines we considered solved these tasks well. **SORL always achieved _at least_ the second-best performance in _every_ environment and achieved the _best_ performance on 5 environments (25 tasks).** On the two environments where SORL does not achieve the best performance among the baselines, there is only one algorithm that outperforms it (ReBRAC on antmaze-giant and IFQL on humanoidmaze-large). Both ReBRAC and IFQL fail to achieve the best performance on any other environment, highlighting the difficulty of having reaching optimal performance on every environment.
> - At a higher-level, the antmaze-giant and humanoidmaze-large tasks are known to be challenging, long-horizon environments, and it is likely that these environments would benefit significantly from online exploration. As noted in the limitations discussed in Appendix B, SORL does not have a principled exploration strategy, which is standard for _offline_ algorithms. Depending on the coverage and quality of the offline dataset, purely offline learning may get stuck in local optima.
> - Relatedly, it is possible that policy extraction is not the primary bottleneck in these two environments under the given dataset size (1 million transitions). It may be the case that, for these two environments, **value learning is challenging**. If the learned value function (e.g. Q function) is poor, then it will be difficult to optimize a policy under that value function, irrespective of the policy extraction method, expressivity of the policy class, or discretization/integration error.
>
> # 5. Computational Resources During Inference
> > While SORL aims to improve training efficiency, what is the trade-off in terms of computational resources required during inference? Does the potential for test-time scaling compensate for any increased computational demands?
> - SORL offers flexibility during both training and inference. That is, SORL can take advantage of a larger compute budget during both training and/or inference. In Figure 2, we show that **SORL can compensate for less training-time computation with greater test-time computation**. Specifically, we consider a small training compute budget of $M^{\text{BTT}}=1$, indicating that we only use one step of backpropagation through time during training. The baseline performance of SORL under such a small training budget is low ($\text{SuccessRate} = 13$ with a single inference step, $M^{\text{inf}}=1$). However, by applying inference-time sequential scaling (i.e. using inference steps $M^{\text{inf}}=2$) and parallel scaling (i.e. Best-of-N with $N=8$), SORL’s performance improves to $\text{SuccessRate=55}$, top performance among the 10 baselines.
>
> # 6. Comparison To Recent Advances
> > How does SORL compare to other recent advances in offline RL, particularly those that also focus on scaling or improving inference efficiency? Are there any key works that should be included in the discussion to provide a more comprehensive comparison?
> - **We will add the following paragraph to our related work for additional comparison**:
>      - Independent of generative models, prior work has proposed applying a form of best-of-N sampling to actions from the behavior policy (i.e. behavioral candidates) [Chen et al., 2022, Fujimoto et al., 2019, Ghasemipour et al., 2021, Hansen-Estruch et al., 2023, Park et al., 2024b]. Park et al. [2024b] proposed two methods of test-time policy improvement, by using the gradient of the Q-function. One of Park et al. [2024b]’s approaches relies on leveraging test-time states, which SORL’s parallel scaling method does not require. The second approach proposed by Park et al. [2024b] adjusts actions using the gradient of the learned Q-function, which is conceptually similar to our approach of best-of-N sampling with the Q-function verifier. However, their method requires an additional hyperparameter to tune the update magnitude in gradient space.
>
> # 7. Limitations
> > What are the potential limitations of using shortcut models in SORL, especially in terms of capturing long-term dependencies or handling highly complex data distributions?
> - **SORL utilizes a highly expressive policy class, shortcut models, so we believe that SORL is capable of handling highly complex data distributions as well as or better than other generative model-based offline RL methods.** However, the problem of capturing long-term dependencies relies on the Q function, for which we use the standard Bellman error minimization. Learning a robust value model is thought to be challenging in offline RL [Levine et al., 2020], though recent work has argued that policy extraction is the primary bottleneck in offline RL [Park et al., 2024a]. **Our paper’s approach does not improve the value learning component and instead focuses on improving policy learning and inference-time scaling.**
> - Additionally, **like all _offline_ RL algorithms, SORL does not have a principled exploration strategy**, as exploration is generally not necessary or desired in the offline setting. However, if SORL were to be applied in the _online_ setting, the lack of an exploration mechanism may limit SORL's performance on tasks requiring significant exploration.

---

> > ### Comment · Reviewer_GJHL · 2025-08-05
> >
> > I thank the authors for their response that have largely addressed my concerns. Please improve the presentation in the final version,  and I will maintain my current score.

---

### Note · Authors · 2025-08-15

We’d again like to thank the reviewers for their feedback. We summarize our responses to the main comments below.

1. **Evaluation Scope & Robustness** – SORL was evaluated on 40 tasks across 8 environments, including long-horizon, goal-directed planning and low-level motor control, and tasks comprised of stitching sub-tasks. SORL achieves the best or second-best performance on all of the environments, including the *top* performance on the majority of environments. In response to Reviewers GJHL, VRv3, and Ewda’s questions, we provided a more thorough analysis of SORL’s performance and the factors that affected its performance.
2. **Theoretical Novelty** – We provide, to the best of our knowledge, the first theoretical analysis of shortcut models. While the aim of Theorem 2 aligns with standard offline RL theory, its application to shortcut models, including integration with inference-time scaling, presents a novel contribution.
3. **Clarity & Corrections** – We fixed the typos and visual improvements suggested by the reviewers.
4. **Related Work** – In response to Reviewer rsZE’s question, we expanded the related work to include recent work on improving inference efficiency in RL.

We thank the reviewers for their engagement and for suggestions that will strengthen the final version of the paper.

---

### Decision · Program_Chairs · 2025-09-17

**Decision:**

Accept (poster)

**Comment:**

This paper proposes SORL, an offline RL algorithm that integrates shortcut models with flow-matching, Q-learning, and inference-time scaling. The reviewers generally agree that the paper is clearly written, well-motivated, and supported by a broad set of experiments. They find the approach timely and relevant, with strong empirical results and a compelling evaluation.

That said, the AC finds the novelty to be limited. The use of generative models (e.g., diffusion and flow matching) to define more expressive policy classes than Gaussian-based policies was pioneered by earlier work such as Diffusion QL. Likewise, diffusion distillation techniques are not new, and one-step training (as opposed to pretraining followed by distillation) has already been studied in the vision domain. The combination of Q-learning loss, diffusion/flow-matching objectives, and shortcut models therefore does not constitute a fundamentally new direction, though applying this combination to offline RL is a useful and relevant contribution.

The AC also has concerns regarding the authors’ claim that “The key advantage of SORL over distillation-based generative model approaches (e.g., SRPO, CAC, FQL) is SORL’s one-stage policy training procedure, as compared to the two-stage distillation-based approaches over which error compounds.” In the vision domain, state-of-the-art one-step generation performance (for unconditional, label-conditional, and text-to-image diffusion models) remains firmly achieved by score-distillation methods, not by training one-step models from scratch. If the authors’ explanation were fully correct, this would contradict established trends in vision research. The AC therefore encourages the authors to revisit this claim and provide a more careful justification for why SORL shows stronger empirical performance than existing pretraining-plus-distillation approaches.

In summary, while the paper does not offer substantial theoretical or algorithmic novelty, it tackles an important and timely problem and demonstrates strong empirical results. On balance, the AC considers this a relevant contribution to the offline RL community, but advises the authors to moderate claims about the source of performance gains in the final version.